# Approximate optimization of convex functions with outlier noise

**Anindya De**
University of Pennsylvania
anindyad@cis.upenn.edu

**Sanjeev Khanna**
University of Pennsylvania
sanjeev@cis.upenn.edu

**Huan Li**
University of Pennsylvania
huanli@cis.upenn.edu

**Hesam Nikpey**
University of Pennsylvania
hesam@cis.upenn.edu

## Abstract

We study the problem of minimizing a convex function given by a zeroth order oracle that is possibly corrupted by *outlier noise*. Specifically, we assume the function values at some points of the domain are corrupted arbitrarily by an adversary, with the only restriction being that the total volume of corrupted points is bounded. The goal then is to find a point close to the function's minimizer using access to the corrupted oracle.

We first prove a lower bound result showing that, somewhat surprisingly, one cannot hope to approximate the minimizer *nearly as well* as one might expect, even if one is allowed *an unbounded number* of queries to the oracle. Complementing this negative result, we then develop an efficient algorithm that outputs a point close to the minimizer of the convex function, where the specific distance matches *exactly*, up to constant factors, the distance bound shown in our lower bound result.

## 1 Introduction

Unconstrained convex optimization is among the most well-studied problems in mathematical optimization and has extensive applications in machine learning [7]. In the classic unconstrained convex minimization problem, we are given oracle access to a convex function $f : \mathbb{R}^d \to \mathbb{R}$, and seek to efficiently find a point $\tilde{x}$ that is close to the minimizer of $f$ (call it $x^*$)[1]. To obtain meaningful guarantees for approximating the minimizer $x^*$, one needs to make certain assumptions on the convexity and smoothness of $f$. In particular, $f$ is commonly assumed to be $\alpha$-strongly convex and $\beta$-smooth for some $\beta > \alpha > 0$. This means that, when $f$ is twice differentiable, its second derivative in any direction is between $\alpha$ and $\beta$. For ease of presentation, we will say a convex function is $(\alpha, \beta)$-*nice* if it satisfies these two conditions.

It is well known that the minimizer of an $(\alpha, \beta)$-nice function can be approximated arbitrarily well in polynomial time by the classic gradient descent algorithm, or its accelerated version due to Nesterov [18]. To formally state the performance of these two algorithms, let us suppose an $(\alpha, \beta)$-nice function $f : \mathbb{R}^d \to \mathbb{R}$ is given to us as a *zeroth order oracle*, which returns the function value $f(x)$ on any input point $x \in \mathbb{R}^d$. Then we have:

**Theorem 1.1** ([7]). *Given any initial point $x_0 \in \mathbb{R}^d$ and $\epsilon > 0$, the classic gradient descent algorithm outputs a point $\tilde{x}$ s.t. $\|\tilde{x} - x^*\|_2 \leq \epsilon$ using $O(d(\beta/\alpha) \log \frac{\|x_0 - x^*\|}{\epsilon})$ oracle queries.*

---

[1]Sometimes, instead of finding a point close to $x^*$, the goal is to find a point whose *function value* is close to $f(x^*)$. However, as we point out later, it is more natural to study approximations in the domain in our setting.

35th Conference on Neural Information Processing Systems (NeurIPS 2021).

**Theorem 1.2** ([18])**.** *Given any initial point $x_0 \in \mathbb{R}^d$ and $\epsilon > 0$, the accelerated gradient descent algorithm outputs a point $\tilde{x}$ s.t. $\|\tilde{x} - x^*\|_2 \leq \epsilon$ using $O(d\sqrt{\beta/\alpha} \log \frac{\|x_0 - x^*\|}{\epsilon})$ oracle queries.*

We note that the quantity $\beta/\alpha$ is called the *condition number* of the function $f$. The efficiency of a convex minimization algorithm is usually measured with respect to such a quantity.

Given that the complexity of convex minimization with *exact* oracles has been well understood, a natural question is "can we still minimize a convex function efficiently if the oracle is *to some extent inaccurate*?" There has in fact been a rich body of work in efforts to answer this question. Roughly, they can be divided into two categories based on their assumptions on the oracle's inaccuracy.

The first category studies the case when the oracle is corrupted by *stochastic noise*[2] — namely, the errors of the oracle are assumed to be random and independently drawn from some distribution [10, 20, 19]. Notice that as the oracle is correct in expectation, one can obtain good estimates to the true values efficiently by averaging over sufficiently many points in a small neighborhood. As a result, it is still possible to approximate the minimizer $x^*$ to within arbitrarily small distance in polynomial time. The main focus of these results is thus to obtain optimal algorithm efficiency, ideally matching that of Nesterov's accelerated gradient descent.

The second category, on the other hand, considers the *adversarial noise*. That is, the errors are no longer drawn from certain distributions, but rather are added adversarially, while obeying certain constraints. A representative such model is what we call *pointwise-bounded noise model* [22, 5], in which the only assumption on the noise is that it is pointwise bounded in magnitude; other than that, the specific perturbation on each point can be arbitrary. Formally, it is allowed for the oracle to return, on an input point $x$, a value in the range $[f(x) - \epsilon, f(x) + \epsilon]$ *(absolute errors)* or $[(1 - \epsilon)f(x), (1 + \epsilon)f(x)]$ *(relative errors)*, for some $\epsilon > 0$. [5] shows that when the errors are absolute and $\epsilon$ is on the order of $1/d$, there is a polynomial-time algorithm that can find a point with function value arbitrarily close to $f(x^*)$. Whereas [22] shows that, in sharp contrast, when $\epsilon$ is about $1/\sqrt{d}$ or larger, no polynomial-time algorithm is able to find a point with function value within some constant error of $f(x^*)$, for both absolute and relative errors.

**Our model.** Following the second line of research above, in this paper we study another type of adversarial noise model, called *outlier noise model*, which can be seen as a natural variant of the pointwise-bounded noise model in [22, 5]. In this model, the only assumption we make is a bound on the "number" of points corrupted by the noise; apart from that, we do *not* assume any bounds on the magnitude of the errors on the corrupted points. Formally, we are given access to an exact zeroth order oracle of a function $\hat{f}$ that differs from the true convex function $f$ *only* on a set $\mathcal{C} \subset \mathbb{R}^d$, with the guarantee that the volume of $\mathcal{C}$ is at most that of a $d$-dimensional ball of radius $K$, for some $K > 0$. Notice that although $\mathcal{C}$ is bounded in volume, for any $x \in \mathcal{C}$, $\hat{f}(x)$ and $f(x)$ can differ *arbitrarily*.

Though variants of each other, both the pointwise-bounded noise model in [22, 5] and our model may be considered as special cases of a more general type of adversarial noise model, namely the $\ell_p$-*bounded noise model*. To see this, let us write the noise as a function $\eta : \mathbb{R}^d \to \mathbb{R}$, such that the noisy zeroth order oracle given to us corresponds to the function $f + \eta$. Then, in the (absolute) pointwise-bounded noise model, $\eta$ is bounded in $\ell_\infty$-norm (they assume $\|\eta\|_\infty \leq \epsilon$), whereas in the outlier noise model, $\eta$ is bounded in $\ell_0$-norm (we assume $\|\eta\|_0 \leq \texttt{vol}(\text{radius-}K \text{ ball})$). It will certainly be an interesting future direction to explore what can be achieved for minimizing convex functions with general $\ell_p$-bounded noise.

We would also like to point out that in our noise model, it is more natural to consider getting as close as possible to $x^*$ as opposed to finding a point with function value close to $f(x^*)$. This is because our only assumption on the noise is that the corrupted region $\mathcal{C}$, which is in the *domain* of the function $f$, is bounded in volume, and in particular, it may be located around $x^*$. Therefore, we believe it makes the most sense to measure the quality of the solution of a minimization algorithm by its distance to the optimal in the domain (specifically, how the distance compares with the corruption radius $K$).

**Our results.** Let us first consider what one might expect to achieve in the outlier noise model. By the observation made above, it is not hard to see that we cannot hope to always find a point that is within distance $K$ of the minimizer $x^*$, as an adversary could potentially corrupt some radius-$K$ ball

---

[2]This model more often deals with the *first order oracle*, which gives the (noisy) gradient $\nabla f(x)$ at a point $x$.

around $x^*$, making it impossible for us to know where the true minimizer lies. However, is it possible to obtain a distance bound that is close to $K$ (e.g. $O(K)$)?

Our first result in this paper is a lower bound showing that, somewhat surprisingly, this goal is in general impossible to achieve, even for algorithms that are allowed *an unbounded number* of queries to the oracle. In fact, our lower bound indicates that the best distance bound one can hope for is at least $\Omega(K\sqrt{\beta/\alpha})$, where we recall that $\alpha, \beta$ are the "niceness" of the function. This result is a consequence of the existence of two $(\alpha, \beta)$-nice functions that only differ in a small region of the domain, but whose minimizers are sufficiently far apart. We then develop, as our second result, an efficient algorithm that finds a point within distance $O(K\sqrt{\beta/\alpha})$ of the minimizer $x^*$, thus matching our lower bound up to constant factors. Roughly, our algorithm performs two stages of gradient descent, using approximate gradients computed from the noisy zeroth order oracle.

We state our results formally in the theorems below. For simplicity, let us say a zeroth order oracle of a function $f$ is *$K$-corrupted* if it is perturbed by an outlier noise of corruption radius $K$.

**Theorem 1.3** (Lower bound; formal version appears as Theorem 3.1). *For $\alpha, \beta, K > 0$ and sufficiently large $d$, there exist two $(\alpha, \beta)$-nice functions $f_0, f_1 : \mathbb{R}^d \to \mathbb{R}$ which satisfy the following conditions: (i) The $\ell_2$-distance between the minimizers of $f_0, f_1$ is $\Omega(K\sqrt{\beta/\alpha})$; (ii) The total volume of points where $f_0$ and $f_1$ differ is at most that of a $d$-dimensional ball of radius $K$.*

To see the implication of this theorem, consider an adversary that randomly picks an index $i \in \{0, 1\}$ and lets $f_i$ be the true convex function, but *always* uses $f_0$ as the $K$-corrupted oracle. Then no matter which point our algorithm outputs, with probability at least $1/2$ over the randomness of $i$, it is $\Omega(K\sqrt{\beta/\alpha})$ away from the minimizer of the true convex function $f_i$.

**Theorem 1.4** (Algorithm; formal version appears as Theorems 4.2, 5.1). *Let $\alpha, \beta, K > 0$ and $d$ be sufficiently large. There is an algorithm that, given access to a $K$-corrupted oracle of an $(\alpha, \beta)$-nice function $f : \mathbb{R}^d \to \mathbb{R}$ and an initial point $x_0 \in \mathbb{R}^d$, makes $\tilde{O}(d \cdot (\beta^2/\alpha^2) \cdot \log \|x_0 - x^*\|)^3$ queries to the oracle and outputs a point $\hat{x}$ s.t. $\|\hat{x} - x^*\|_2 \leq O(K\sqrt{\beta/\alpha})$. Here $x^*$ is the minimizer of $f$.*

We note that in both our results, our conditions on $d$ are roughly that $d \geq \Omega(\log \beta/\alpha)$. In other words, we require the function's condition number $\beta/\alpha$ to be at most $2^{O(d)}$, a quantity exponential in $d$.

**Our techniques.** We now briefly describe our techniques used to obtain the above results.

Our lower bound proceeds by proving the existence of two $(\alpha, \beta)$-nice functions $f_0$ and $f_1$ s.t. (i) they differ only in an ellipsoid with volume at most that of a ball of radius $K$; (ii) their minimizers are at $\ell_2$-distance $\Omega(K\sqrt{\beta/\alpha})$-away from each other. As mentioned above, this implies that even with infinitely many queries, one cannot approximate the minimizer $x^*$ to within distance $o(K\sqrt{\beta/\alpha})$.

More concretely, we start by letting $f_0$ be a simple $(\alpha, \beta)$-nice quadratic function whose minimizer is at the origin. Then the existence of our desired function $f_1$ will follow by two steps: first we show that there exists another function $f'$ with certain good properties, then we obtain $f_1$ by taking a piecewise combination of $f_0$ and $f'$. Specifically, the properties that we need $f'$ to satisfy is as follows: (i) $f'$ is $(\alpha, \beta)$-nice; (ii) both the function values and the gradients of $f_0$ and $f'$ agree on all points on the periphery of an ellipsoid $\mathcal{E}$ that is centered at the origin and has bounded volume; (iii) the minimizer of $f'$ has coordinate of the form $(\Omega(K\sqrt{\beta/\alpha}), 0, \ldots, 0)$. We show that this essentially boils down to a convex function interpolation task where the function values and gradients at some *infinitely* many points are given. To this end, we adapt an interpolation result from [23], which was proposed to accommodate interpolation from *finitely* many points, to prove the existence of $f'$. Finally, we construct the function $f_1$ by letting $f_1 = f'$ inside the ellipsoid $\mathcal{E}$, but $f_1 = f_0$ outside of $\mathcal{E}$.

For the upper bound, our algorithm proceeds in two stages. In the first stage, we come to within distance $O(K(\beta/\alpha))$ of the minimizer $x^*$ and in the subsequent stage, we improve it to $O(K\sqrt{\beta/\alpha})$.

The first algorithm essentially follows a gradient descent, but using an approximate gradient at each step. In order to estimate the gradient at a point $x$, we set up a system of linear equations, where each equation adds a constraint on the derivative at $x$ along a uniformly random direction. There are however two types of noise in these equations, one from using a zeroth order oracle to compute first

---

[3]Here and throughout the paper, we write $\tilde{O}(f)$ to denote $O(f \cdot \text{poly}(\log f))$.

order information, and the other from the outlier noise added to the oracle. While we can solve this noisy linear system with good enough accuracy via an exhaustive search, we show that using an LP decoding routine in [12], we can solve it more efficiently in polynomial time. We note that using the latter is only to improve the running time, as both approaches result in the same query complexity.

However, the one bottleneck in this approach is that at any point, the ball of radius $< K$ around it could potentially be (nearly) completely corrupted. Thus, to get a meaningful estimate of the gradient, we have to sample points which are more than distance $K$-far apart. This tradeoff eventually allows us to get $O(K(\beta/\alpha))$-close to the minimizer.

To get to the optimal closeness of $O(K\sqrt{\beta/\alpha})$, we next start at a point which is guaranteed to be $O(K(\beta/\alpha))$-close to the true minimizer. We now consider the function $\bar{f}$ which is defined as the average of $f$ in a ball of radius (say) $2K$. It is not hard to verify that $\bar{f}$ continues to be $(\alpha, \beta)$-nice. Moreover, the minimizers of $\bar{f}$ and $f$ can be shown to be $O(K\sqrt{\beta/\alpha})$-close to each other. Thus, it suffices to get close to the minimizer of $\bar{f}$. We will do so by simulating a gradient descent on $\bar{f}$. It therefore boils down to how we can efficiently approximate the gradients of $\bar{f}$.

By the definition of $\bar{f}$, the gradient $\nabla \bar{f}(x)$ is also equal to the average of the gradients $\nabla f(y)$ over all points $y$ within distance $2K$ of $x$. This suggests that we can approximate $\nabla \bar{f}(x)$ by averaging the gradients $\nabla f(y)$ at sufficiently many randomly sampled $y$'s, and bounding the error using concentration inequalities for sums of random vectors. Now a key observation is that, since these $y$'s are sampled *randomly* from a radius $2K$-ball, it is highly likely that each sampled $y$ sits in a mostly uncorrupted neighborhood, as long as the dimension is sufficiently high. Consequently, we can use the LP decoding approach above to obtain an accurate estimate of the gradient at each of these points.

**Prior work on noisy convex optimization.** Other than the works mentioned above, noisy convex optimization has also been investigated in the context of multi-armed bandits and regret minimization [2, 8, 1]. In the direction of convex optimization under adversarial noise, the early results in fact date back to the 90s by [3]. Specifically for the pointwise-bounded noise model, there are subsequent works such as [21, 24, 17] that have improved on the guarantees of [5].

Due to space limitation, we include some other related work in Appendix A.

**Organization.** In Section 2, we set up a few notations and give some basic definitions and technical preliminaries. In Section 3, we prove our lower bound result. In Section 4, we give a first algorithm that gets us to within distance $O(K(\beta/\alpha))$ of $x^*$. In Section 5, we give a second algorithm that gets us to within distance $O(K\sqrt{\beta/\alpha})$ of $x^*$. In Section 6, we propose several future directions.

## 2 Preliminaries

Note that due to space limitation, we defer some of the preliminaries to Appendix B.

While there are many known equivalent definitions of strong convexity and smoothness of a function, the specific ones that we use in this paper are as follows.

**Definition 2.1.** A function $f : \mathbb{R}^d \to \mathbb{R}$ is $\beta$-smooth if it is differentiable and for all pairs $x, y \in \mathbb{R}^d$ we have $\|\nabla f(x) - \nabla f(y)\| \leq \beta \|x - y\|$.[4]

**Definition 2.2.** A function $f : \mathbb{R}^d \to \mathbb{R}$ is $\alpha$-strongly convex if $f(x) - \frac{\alpha}{2} \|x\|^2$ is *convex*.

We next set up notations of a ball and the uniform distribution over it.

**Definition 2.3.** Let $\mathcal{B}(x, r) \stackrel{\text{def}}{=} \{y : \|y - x\| \leq r\}$ denote the ball of radius $r$ centered at $x$. Let $\mathcal{U}(x, r)$ denote the uniform distribution over all points in the ball $\mathcal{B}(x, r)$.

As a result, the fraction of corrupted volume in a ball $\mathcal{B}(x, r)$ is $\Pr_{y \sim \mathcal{U}(x,r)}[f(y) \neq \hat{f}(y)]$, where we recall that $\hat{f}$ denotes the corrupted version of $f$.

For our second algorithm in Section 5, we will need to consider the "average" function, whose value at a point $x$ is the average of $f(y)$'s where $y$ is within some distance of $x$.

---

[4]Here and going forward, all norms are $\ell_2$-norms unless stated otherwise

**Definition 2.4.** For any $r > 0$, define the function $\bar{f}_r$ as $\bar{f}_r(x) \stackrel{\text{def}}{=} \mathbb{E}_{y \sim \mathcal{U}(x,r)}[f(y)]$.

It is not hard to verify the strong convexity and smoothness of $\bar{f}_r$:

**Lemma 2.5.** *If $f$ is $\alpha$-strongly convex and $\beta$-smooth, $\bar{f}_r$ is also $\alpha$-strongly convex and $\beta$-smooth.*

As a result of $\alpha$-strong convexity and $\beta$-smoothness, we can upper bound the distance between the minimizers of $f$ and $\bar{f}_r$ by $O(r\sqrt{\beta/\alpha})$.

**Lemma 2.6.** *Let $x^*, \bar{x}_r$ be the minimizers of $f$ and $\bar{f}_r$ respectively. Then $\|x^* - \bar{x}_r\| \leq 2r\sqrt{\beta/\alpha}$.*

A proof of this lemma is included in Appendix B.

# 3   An $\Omega(K\sqrt{\beta/\alpha})$ lower bound

In this section we show that getting $O(K\sqrt{\beta/\alpha})$-close to $x^*$ is the best we can hope for even if we are allowed to query the function value at every point of the domain. We will prove this by showing that when the dimension is sufficiently high in terms of $\beta/\alpha$, there exist two $\alpha$-strongly convex, $\beta$-smooth functions that differ only in an ellipsoid of volume equal to a ball of radius $K$, but whose minimizers are $\Omega(K\sqrt{\beta/\alpha})$-apart.

**Theorem 3.1.** *Given $0 < \alpha \leq \beta$ with $1 + \log \frac{\beta}{\alpha} \leq d$ where $d$ is the dimension, and a $K > 0$, there exist two $\alpha$-strongly convex, $\beta$-smooth functions whose values differ only in an ellipsoid of volume equal to a radius-$K$ ball, but whose minimizers are $\Omega(\sqrt{\frac{\beta}{\alpha}}K)$-far from each other.*

In order to prove Theorem 3.1, we shall prove several intermediate lemmas first, which are built on the interpolation results from [23]. We remark that the main results in [23] are stated for interpolating a set of finitely many points, while for our purpose we need to interpolate infinitely many. Therefore we cannot use their results directly in a black-box manner, but instead have to make certain adaptations.

## 3.1   Some interpolation results from [23]

First let us define the notion of $(\alpha, \beta)$-interpolability.

**Definition 3.2** ($(\alpha, \beta)$-interpolability)**.** Suppose we are given a set of (possibly infinitely many) tuples $\{(x_i, g_i, f_i)\}_{i \in I}$ where each $x_i, g_i \in \mathbb{R}^d$, $f_i \in \mathbb{R}$. Let $\alpha \in \mathbb{R}_{\geq 0}, \beta \in \mathbb{R}_{\geq 0} \cup \{+\infty\}$ where $\alpha < \beta$. We say this set is $(\alpha, \beta)$-interpolable if there is a proper and closed convex function $f : \mathbb{R}^d \to \mathbb{R} \cup \{+\infty\}$ that is $\alpha$-strongly convex and $\beta$-smooth such that for all $i \in I$, $g_i \in \partial f(x_i)$ and $f(x_i) = f_i$, where $\partial f(x_i)$ denotes the set of subgradients of $f$ at $x_i$.

Note here that when $\alpha = 0$, we only require $f$ to be convex. When $\beta = \infty$, we do not require $f$ to be smooth and thus $f$ is not necessarily differentiable; when $\beta < \infty$, the condition $g_i \in \partial f(x_i)$ is equivalent to $g_i = \nabla f(x_i)$ as the gradient is unique at any point when $f$ is differentiable.

The following two lemmas are proved in [23]. The first lemma enables us to reduce the $(\alpha, \beta)$-interpolation of some tuple set to the $(0, \beta')$-interpolation of another tuple set, while the second lemma allows us to further reduce it to the $(\alpha', \infty)$-interpolation of some other tuple set. We note that although [23] only states these lemmas for sets containing finitely many tuples, their proofs work for sets containing infinitely many tuples as well.

**Lemma 3.3.** *Given a set of (possibly infinitely many) tuples $\{(x_i, g_i, f_i)\}_{i \in I}$ where $x_i, g_i \in \mathbb{R}^d$, $f_i \in \mathbb{R}$ and $0 \leq \alpha < \beta \leq +\infty$. The following two statements are equivalent:*

  *1.  $\{(x_i, g_i, f_i)\}_{i \in I}$ is $(\alpha, \beta)$-interpolable.*

  *2.  $\left\{\left(x_i, g_i - \alpha x_i, f_i - \frac{\alpha}{2}\|x_i\|^2\right)\right\}_{i \in I}$ is $(0, \beta - \alpha)$-interpolable.*

**Lemma 3.4.** *Given a set of (possibly infinitely many) tuples $\{(x_i, g_i, f_i)\}_{i \in I}$ where $x_i, g_i \in \mathbb{R}^d$, $f_i \in \mathbb{R}$ and $0 < \beta \leq +\infty$. The following two statements are equivalent:*

  *1.  $\{(x_i, g_i, f_i)\}_{i \in I}$ is $(0, \beta)$-interpolable.*

2. $\left\{(g_i, x_i, x_i^\top g_i - f_i)\right\}_{i \in I}$ is $(1/\beta, \infty)$-*interpolable.*

Then as in [23], by alternately applying Lemmas 3.3 and 3.4 twice each, we are able to reduce any $(\alpha, \beta)$-interpolation problem to a $(0, \infty)$-interpolation problem, where we only want to interpolate some points with a proper and closed *convex* function. Formally, we have the following lemma, whose proof is deferred to Appendix C.

**Lemma 3.5.** *Given a set of (possibly infinitely many) tuples* $\{(x_i, g_i, f_i)\}_{i \in I}$ *where* $x_i, g_i \in \mathbb{R}^d$, $f_i \in \mathbb{R}$ *and* $0 \le \alpha < \beta \le +\infty$. *The following two statements are equivalent:*

1. $\{(x_i, g_i, f_i)\}_{i \in I}$ *is* $(\alpha, \beta)$-*interpolable.*

2. $\left\{\left(\frac{\beta x_i}{\beta - \alpha} - \frac{g_i}{\beta - \alpha}, g_i - \alpha x_i, \frac{\alpha x_i^\top g_i}{\beta - \alpha} + f_i - \frac{\beta \alpha \|x_i\|^2}{2(\beta - \alpha)} - \frac{\|g_i\|^2}{2(\beta - \alpha)}\right)\right\}_{i \in I}$ *is* $(0, \infty)$-*interpolable.*

## 3.2 Our lower bound

We first show that there exists an $\Omega(1)$-strongly convex, $O(1)$-smooth function whose minimizer is $1/2$-far from the origin, but whose function values and gradients agree with the quadratic function $\|x\|^2$ on all points on the surface of a unit ball. Formally, we have the following lemma. For ease of presentation, let us define $X_{=1} \overset{\text{def}}{=} \{x : \|x\| = 1\}$ and similarly $X_{\ge 1} \overset{\text{def}}{=} \{x : \|x\| \ge 1\}$. We also write $e_1 = (1, 0, \ldots, 0)^T$ to denote the first standard basis vector.

**Lemma 3.6.** *Let* $f(x) \overset{\text{def}}{=} \|x\|^2$ *which is* 2-*strongly convex and* 2-*smooth. There exists a* $\frac{1}{2}$-*strongly convex,* 16-*smooth function* $\tilde{f}$ *such that*

1. $\tilde{f}$'s *minimizer is* $\frac{1}{2}e_1$.

2. *For all* $x \in X_{=1}$ *we have* $\tilde{f}(x) = f(x)$ *and* $\nabla \tilde{f}(x) = \nabla f(x)$.

The proof of this lemma is deferred to Appendix C. Roughly, the proof consists of three steps: (i) formulate proving the existence of $\tilde{f}$ as a $(\frac{1}{2}, 16)$-interpolation problem; (ii) use Lemma 3.5 to reduce it to the $(0, \infty)$-interpolation of some infinitely many points; (iii) explicitly construct a proper and closed convex function that does interpolate these points.

Now by taking a piecewise combination of the function $\tilde{f}$ in Lemma 3.6 and the quadratic function $\|x\|^2$, we can show that there exists an $\Omega(1)$-strongly convex, $O(1)$-smooth function $\hat{f}$ whose minimizer is $1/2$-far from the origin, but whose function values and gradients agree with $\|x\|^2$ on every point with $\ell_2$-norm greater than or equal to 1.

**Lemma 3.7.** *Let* $f(x) \overset{\text{def}}{=} \|x\|^2$ *which is* 2-*strongly convex and* 2-*smooth. Define* $\hat{f}$ *such that* $\hat{f}(x) = \tilde{f}(x)$ *if* $\|x\| \le 1$ *and* $\hat{f}(x) = f(x)$ *otherwise* $(\|x\| > 1)$. *Then we have*

1. $\hat{f}$ *is* $\frac{1}{2}$-*strongly convex and* 16-*smooth.*

2. $\hat{f}$'s *minimizer is* $\frac{1}{2}e_1$.

3. *For all* $x \in X_{\ge 1}$ *we have* $\hat{f}(x) = f(x)$ *and* $\nabla \hat{f}(x) = \nabla f(x)$.

The proof of this lemma is included in Appendix C.

Then by scaling the domains of $f, \hat{f}$ in Lemma 3.7, we prove that for any $\kappa \ge 1$, when the dimension is sufficiently high in terms of $\kappa$, there exist two $\Omega(1/\kappa)$-strongly convex, $O(1)$-smooth functions whose function values and gradients agree on every point outside of an ellipsoid of volume equal to a unit ball, but whose minimizers are $\sqrt{\kappa}/2$-apart. Here $\kappa$ shall be thought of as $\beta/\alpha$ where $\beta = \Theta(1)$.

**Lemma 3.8.** *Given* $\kappa \ge 1$ *with* $1 + \log \kappa \le d$ *where* $d$ *is the dimension. Let* $\gamma \overset{\text{def}}{=} (1/\kappa)^{\frac{1}{d-1}} \in [1/2, 1]$. *Let* $S_{d \times d} = \text{DIAG}(\kappa, \gamma, \ldots, \gamma)$. *Define* $s(x) = x^\top S^{-1} x$, *which is* $(2/\kappa)$-*strongly convex and* $(2/\gamma)$-*smooth. Let* $X_{s \ge 1} = \{x : s(x) \ge 1\}$. *Also define* $\hat{s}(x) = \hat{f}(S^{-1/2} x)$. *Then we have*

1. $\hat{s}$ *is* $1/(2\kappa)$-*strongly convex and* $(16/\gamma)$-*smooth.*

2. $\hat{s}$'s minimizer is $\frac{\sqrt{\kappa}}{2}e_1$.

3. For all $x \in X_{s \geq 1}$ we have $\hat{s}(x) = s(x)$ and $\nabla \hat{s}(x) = \nabla s(x)$.

Finally, by further scaling (the domain and the function values of) $s, \hat{s}$ in Lemma 3.8, we can prove Theorem 3.1. The proofs of Lemma 3.8 and Theorem 3.1 are both included in Appendix C.

## 4 An $O(K(\beta/\alpha))$-close algorithm

In this section we give an algorithm GDSTAGEI that finds a point $O(K(\beta/\alpha))$-close to the minimizer of $f$. GDSTAGEI essentially implements a gradient descent algorithm, but uses approximate gradient computed from the noisy oracle at each step. To begin with, we present a subroutine GRADIENTCOMP for computing the gradient at a point where a small neighborhood is mostly uncorrupted.

---

**Algorithm 1:** GRADIENTCOMP($\hat{f}, x, \beta, \tau$)

---

**Input** : $\hat{f} : \mathbb{R}^d \to \mathbb{R}$, $x \in \mathbb{R}^d$, $\beta > 0$, and $\tau > 0$.
**Output :** $g \in \mathbb{R}^d$.

1 Randomly choose $1000d$ pairs of points $a_1, b_1 \ldots, a_{1000d}, b_{1000d}$ in the ball $\mathcal{B}(x, \tau)$.
2 Query the function values $\hat{f}(a_j), \hat{f}(b_j)$ for all $j = 1, 2, \ldots, 1000d$.
3 Let $g \in \mathbb{R}^d$ be any vector such that, for at least $800d$ of the $j$'s, the following holds:

$$\frac{\left| g^\top (b_j - a_j) - \left( \hat{f}(b_j) - \hat{f}(a_j) \right) \right|}{\|b_j - a_j\|} \leq \beta\tau. \tag{1}$$

If no such $g$ exists, set $g$ to be an arbitrary vector.

---

We summarize the performance of GRADIENTCOMP below, with the proof deferred to Appendix D. Essentially, the error in the gradient computed by GRADIENTCOMP tends to zero as $\tau \to 0$.

**Lemma 4.1.** *Fix $d > 0$ and $\beta > 0$. There exists a function $\mathtt{err}(\tau)$ satisfying $\lim_{\tau \to 0^+} \mathtt{err}(\tau) = 0$ such that the following holds. Fix any $x \in \mathbb{R}^d$ and $\tau > 0$ such that the radius-$\tau$ ball centered at $x$ is mostly uncorrupted:*

$$\Pr_{y \sim \mathcal{U}(x,\tau)} \left[ f(y) \neq \hat{f}(y) \right] \leq \frac{1}{100}. \tag{2}$$

*Then we have that with probability $1 - 2^{-3d}$, the vector $g$ returned by GRADIENTCOMP satisfies*

$$\|g - \nabla f(x)\| \leq \mathtt{err}(\tau). \tag{3}$$

*The number of queries made by GRADIENTCOMP is $O(d)$.*

We now describe GDSTAGEI in Algorithm 2. Its performance is characterized in Theorem 4.2.

---

**Algorithm 2:** GDSTAGEI($\hat{f}, \alpha, \beta, x_0, R_0, \delta$)

---

**Input** : $\hat{f} : \mathbb{R}^d \to \mathbb{R}$, $0 < \alpha < \beta$, $x_0 \in \mathbb{R}^d$, $R_0 \geq \|x_0 - x^*\|$, and $\delta \in (0, 1)$.
**Output :** $\hat{x} \in \mathbb{R}^d$.

1 Let the iteration count be $T \leftarrow 100\frac{\beta}{\alpha} \log \frac{R_0}{(\beta/\alpha)K}$.
2 **for** $t = 0, 1, \ldots, T - 1$ **do**
3      Let the sample count be $s \leftarrow 200 \log(T/\delta)$.
4      Sample $s$ random points $y_1, y_2, \ldots, y_s$ in the ball $\mathcal{B}(x_t, 99K)$.
5      Compute gradients $g_i \leftarrow$ GRADIENTCOMP($\hat{f}, y_i, \beta, \tau$) for some sufficiently small $\tau > 0$.
6      Find a vector $\hat{g} \in \mathbb{R}^d$ such that at least $(2s)/3$ of the $g_i$'s are within euclidean distance
        $99.5\beta K$ of $\hat{g}$; if no such $\hat{g}$ exists, set $\hat{g}$ to be an arbitrary vector.
7      Perform a descent step: $x_{t+1} \leftarrow x_t - \frac{1}{2\beta}\hat{g}$.

8 **return** $x_t$.

---

**Theorem 4.2.** *Let $d \geq 2$. Given an initial point $x_0$ with $\|x_0 - x^*\| \leq R_0$ and a $\delta \in (0,1)$, the algorithm GDSTAGEI returns a point $\hat{x}$ with $\|\hat{x} - x^*\| \leq 10000(\beta/\alpha)K$ with probability $1 - \delta$, where $x^*$ is the minimizer of $f$. The number of queries made by GDSTAGEI is $\tilde{O}(d(\beta/\alpha) \log \frac{R_0}{(\beta/\alpha)K} \log(1/\delta))$.*

Crucial to proving this theorem is to show that the gradients used by GDSTAGEI are accurate enough:

**Lemma 4.3.** *Let $d \geq 2$. The $\hat{g}$ computed at Line 6 of GDSTAGEI satisfies with probability $1 - \frac{\delta}{T}$ that $\|\hat{g} - \nabla f(x_t)\| \leq 200\beta K$.*

Full proofs of Theorem 4.2 and Lemma 4.3 are presented in Appendix D.

**A note on the running time of our algorithms.** While we are mainly concerned about the query complexity, we remark that both of our algorithms above can be implemented in polynomial time.

For GRADIENTCOMP, all steps except Line 3 are easily seen to be implementable in polynomial time (in particular, linear time). Therefore it suffices to show that Line 3 can be done efficiently.

**Claim 4.4.** *A vector $g$ satisfying the condition at Line 3 of GRADIENTCOMP, if it exists, can be found in time polynomial in $d$.*

Our proof of this claim proceeds by presenting a $\text{poly}(d)$-time algorithm based on an LP-decoding routine in [12]. Therefore let us first introduce the specific result that we need from [12].

Let $A \in \mathbb{R}^{n \times d}$ be a matrix and $z \in \mathbb{R}^d$ be a vector. Consider the linear system $Ax = Az$, to which $x = z$ is clearly a solution. If $n \geq d$ and $A$ has full rank, then we can retrieve the vector $z$ given $A$ and $Az$ by solving the linear system $Ax = Az$ in polynomial time using, e.g., Gaussian elimination.

Now suppose the RHS of the linear system is corrupted by some noise $e \in \mathbb{R}^n$, and we are *only* given $A$ and the corrupted RHS $Az + e$, then can we still retrieve the vector $z$ efficiently? [12] showed that under certain assumptions, we can obtain good estimates of $z$ in poly-time by linear programming.

**Theorem 4.5** ([12])**.** *There exist constants $\rho^* \approx 0.239$ and $\gamma \geq 1$ such that the following holds. Suppose $n \geq \gamma d$ and $A_{n \times d}$'s entries are drawn independently from a standard Gaussian distribution. Suppose also the noise $e$ can be written as $e = e_1 + e_2$ where $\|e_1\|_0 \leq \rho^* n$. Then given $A$ and $Az + e$, we can find in polynomial time a vector $z'$ s.t. $\|z - z'\|_2 \leq O(\|e_2\|_\infty)$, for any $z \in \mathbb{R}^d$.*

Basically, this theorem assumes that the noise can be decomposed into the sum of two parts, one with small nonzero support, and the other with small entry-wise magnitude. Then the $\ell_2$-error of the solution is on the order of the largest entry-wise magnitude of the second part of the noise.

*Proof of Claim 4.4.* Let us define a matrix $B \in \mathbb{R}^{1000d \times d}$ whose $i^{\text{th}}$ row is equal to $\frac{(b_i - a_i)^T}{\|b_i - a_i\|}$, where $a_i, b_i$'s are the sampled points at Line 1 of GRADIENTCOMP. We also define vectors $b, \hat{b} \in \mathbb{R}^{1000d}$ with $b(i) = \frac{\nabla f(x)^T(b_i - a_i)}{\|b_i - a_i\|}$ and $\hat{b}(i) = \frac{\hat{f}(b_i) - \hat{f}(a_i)}{\|b_i - a_i\|}$, where $x$ is the input point of GRADIENTCOMP. Notice that each $b(i)$ is the inner product of the $i^{\text{th}}$ row of $B$ and $\nabla f(x)$, and therefore we have $B\nabla f(x) = b$. Consequently, given $B$ and $b$ we can retrieve $\nabla f(x)$ by solving the linear system $By = b$ ($y$ are the variables). Thus our goal becomes solving this linear system when only $B$ and $\hat{b}$ (a.k.a. a corrupted version of $b$) are given. While this looks like the task in Theorem 4.5, note that the entries of $B$ are *not* drawn from independent Gaussian distributions, so we will need go a step further.

As the $a_i, b_i$'s are sampled uniformly at random from a ball, each row $\frac{(b_i - a_i)^T}{\|b_i - a_i\|}$ of $B$ is a unit vector with a uniformly random direction. It is well known that a vector with independent standard Gaussian entries also points to a uniformly random direction. In fact, we can sample a $d$-dimensional such vector by a three-step process: (i) sample a unit vector with a random direction; (ii) sample a length $\ell$ from the $\chi^2$-distribution with $d$ degrees of freedom (i.e., the sum of the squares of $d$ independent standard Gaussians); (iii) scale the unit vector by $\sqrt{\ell}$. In light of this, let us generate a diagonal matrix $D \in \mathbb{R}^{1000d \times 1000d}$ such that each $D(i, i)$ is independently sampled as in step (ii). Then we consider the linear system $D^{1/2}By = D^{1/2}b$ to which $y = \nabla f(x)$ is a solution. Notice that we now have that each entry of $D^{1/2}B$ follows a standard Gaussian.

By thinking of $D^{1/2}\hat{b}$ as a corrupted version of $Db$, we then need to show that the noise $e \overset{\text{def}}{=} D^{1/2}(\hat{b} - b)$ can be written as $e_1 + e_2$ such that $\|e_1\|_0 \leq \rho^*(1000d)$ and $\|e_2\|_\infty$ is small. To this end,

we notice that for each $i$ such that both $a_i, b_i$ are uncorrupted, we have by $\beta$-smoothness that

$$\left| b(i) - \hat{b}(i) \right| = \left| \frac{\nabla f(x)^T (b_i - a_i)}{\|b_i - a_i\|} - \frac{\hat{f}(b_i) - \hat{f}(a_i)}{\|b_i - a_i\|} \right| \leq O(\beta\tau), \tag{4}$$

where $\tau$ is the radius of the ball $\mathcal{B}(x, \tau)$ from which $a_i, b_i$'s are sampled. Thus, if $\mathcal{B}(x, \tau)$ is mostly (say 99%) uncorrupted, with probability $1 - \exp(-\Omega(d))$, (4) holds for most (say 90%) of the $i$'s.

Also, by standard Markov's inequality and Chernoff bounds, with probability $1 - \exp(-\Omega(d))$, for most (say 99%) of the $i$'s we have $D(i, i) \leq O(d)$. Combining this with (4), we have for 80% of the $i$'s that $\sqrt{D(i,i)} \left| \hat{b}(i) - b(i) \right| \leq O(\sqrt{d}\beta\tau)$, implying the existence of $e_1, e_2$ s.t. $e = e_1 + e_2$ and $\|e_1\|_0 \leq 0.2(1000d)$, $\|e_2\|_\infty \leq O(\sqrt{d}\beta\tau)$. This means that by Theorem 4.5 we can use linear programming to find a $g$ with $\|g - \nabla f(x)\| \leq O(\sqrt{d}\beta\tau)$, matching the guarantee in Lemma 4.1.

Finally, to address a technicality about the constant $\gamma$ in Theorem 4.5, we note that we can increase the number of sampled $a_i, b_i$ pairs to $\max\{1000d, \gamma d\}$, and the rest of the analysis still follows. $\quad\square$

Then we consider the running time of GDSTAGEI. By Claim 4.4 and straightforward observations, all steps other than Line 6 run in polynomial time. Thus we focus on the efficiency of Line 6.

**Claim 4.6.** *A vector $\hat{g}$ satisfying the condition at Line 6 of* GDSTAGEI*, if it exists, can be found in nearly-linear time in $s$, at the cost of an extra constant factor in the radius of the ball.*

We note that an extra constant factor in the radius of the ball will not affect the final distance to $x^*$ by more than a constant factor. The proof of this claim is deferred to Appendix D. Roughly, the proof proceeds by sampling $\tilde{O}(1)$ points from $g_1, \ldots, g_s$ and checking for each sampled $g_j$ if at least $2/3$ fraction of the total points are within euclidean distance $200\beta K$ of $g_j$. Note that the radius now becomes $200\beta K$ as opposed to $100\beta K$ at Line 6 of GDSTAGEI.

# 5   An $O(K\sqrt{\beta/\alpha})$-close algorithm

In this section we give an algorithm GDSTAGEII that, when given an initial point which is $O(K(\beta/\alpha))$-close to the minimizer $x^*$ of $f$, finds a point that is $O(K\sqrt{\beta/\alpha})$-close to $x^*$. GDSTAGEII basically performs a gradient descent on the average function $\bar{f}_{2K}$ (Definition 2.4).

---

**Algorithm 3:** GDSTAGEII($\hat{f}, \alpha, \beta, x_0$)

**Input** : $\hat{f} : \mathbb{R}^d \to \mathbb{R}$, $0 < \alpha < \beta$, and $x_0 \in \mathbb{R}^d$ with $\|x_0 - x^*\| \leq 10000(\beta/\alpha)K$.
**Output :** $\hat{x} \in \mathbb{R}^d$.

1  Let the iteration count be $T \leftarrow 100\frac{\beta}{\alpha}\log(\frac{\beta}{\alpha} + 1)$.
2  **for** $t = 0, 1, \ldots, T - 1$ **do**
3  $\quad$ Let the sample count be $s \leftarrow 400\frac{\beta}{\alpha}\log(dT)$.
4  $\quad$ Sample $s$ random points $y_1, y_2, \ldots, y_s$ in the ball $\mathcal{B}(x_t, 2K)$.
5  $\quad$ Compute gradients $g_i \leftarrow$ GRADIENTCOMP($\hat{f}, y_i, \beta, \tau$) for sufficiently small $\tau > 0$, and
$\quad$ $\quad$ their average $\bar{g} \leftarrow \frac{1}{s}\sum_{i=1}^{s} g_i$.
6  $\quad$ Perform a descent step: $x_{t+1} \leftarrow x_t - \frac{1}{2\beta}\bar{g}$.
7  **return** $x_t$.

---

The performance of GDSTAGEII is characterized in Theorem 5.1, with the proof in Appendix E. Note that while the success probability in Theorem 5.1 is not arbitrarily large, we can amplify it to any $1 - \delta$ by repeating the algorithm $O(\log(1/\delta))$ times, as we show in Corollary E.2.

**Theorem 5.1.** *Suppose that $d \geq 100\log(\beta/\alpha + 1)$. Then given an initial point $x_0$ that satisfies $\|x_0 - x^*\| \leq 10000(\beta/\alpha)K$,* GDSTAGEII *returns a point $\hat{x}$ with $\|\hat{x} - x^*\| \leq 1000\sqrt{\beta/\alpha}K$ with probability at least $1 - 2^{-d/8}$, where $x^*$ is the minimizer of $f$. The number of queries made by* GDSTAGEII *is $\tilde{O}(d(\beta/\alpha)^2)$. Moreover, the algorithm runs in polynomial time.*

The proof of Theorem 5.1 builds on a lemma showing that the gradients that GDSTAGEII uses are sufficiently precise. The lemma relies on an $\ell_2$-concentration inequality for the sum of random vectors (i.e., the Vector Bernstein Inequality in Theorem E.1). Its proof also appears in Appendix E.

**Lemma 5.2.** *Let* $d \geq 100 \log(\beta/\alpha + 1)$. *The vector* $\bar{g}$ *computed at Line 5 of* GDSTAGEII *satisfies the following with probability at least* $1 - 2^{-d/8}/T$: $\left\| \bar{g} - \nabla \bar{f}_{2K}(x_t) \right\| \leq 16\sqrt{\alpha\beta}K$.

## 6  Future directions

We obtained asymptotically matching upper and lower bounds on how well the minimizer of a convex function can be identified in presence of outlier noise. There are several natural directions for future work. First, while our algorithm's query complexity has essentially the same dependence on $d$ and $\|x_0 - x^*\|$ as Nesterov's accelerated gradient descent, it is still off by a factor of $(\beta/\alpha)^{1.5}$. It will thus be interesting to understand if this remaining performance gap can be eliminated. Also, we note that both our results require the dimension to be sufficiently high. While we believe the high dimension regime is of the most interest, it will be an interesting exercise to understand how these bounds change in the low-dimensional setting. Finally, as pointed out in the introduction, an appealing future direction is to study convex minimization with the more general $\ell_p$-bounded noise.

## Acknowledgments and Disclosure of Funding

We thank the anonymous reviewers for their valuable feedback. This work was supported in part by NSF awards CCF-1763514, CCF-1934876, CCF-2008305, CCF-1910534, CCF-1926872, and CCF-2045128.

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
