# A    Other related work

In addition to the work on noisy convex optimization, the current paper is also thematically related to works in learning theory and complexity where the goal is to reconstruct simple classes of functions under outlier noise. This includes work on reconstruction of low-degree polynomials [4, 14, 15]. In particular, [15] gave an efficient algorithm whose error tolerance matches the information theoretic limits. In addition, recently, [9] achieved similar algorithmic guarantees for functions which are sparse in the Fourier space. While similar in spirit, the model in these works differ from the current paper in one crucial way – namely, while we only put a bound on the volume of the outlier locations, they, in addition, assume that the outlier locations are also uniformly distributed in the domain. At a more technical level, the results in [4, 14, 15, 9] crucially rely on techniques originating from coding theory such as the Goldreich-Levin theorem [13] and the Berlekamp-Welch algorithm [6]. In contrast, the results in the current paper depend on a careful adaptation of gradient descent aimed at making it noise tolerant. Very recently, the authors also studied a set of discrete search problems under outlier noise, where we want to find a target element in an (partially) ordered set that is corrupted at a bounded number of locations [11]. Such search problems may be seen as discrete analogs of the convex minimization problem studied in this paper.

# B    Additional preliminaries and missing proofs from Section 2

The following are two well-known facts about $\alpha$-strongly convex and $\beta$-smooth functions, which give quadratic upper and lower bounds on the function.

**Fact B.1.** *If $f$ is $\beta$-smooth, then for any $x, y \in \mathbb{R}^d$ we have*

$$f(y) \leq f(x) + \nabla f(x)^\top (y - x) + \frac{\beta}{2} \|y - x\|^2 .$$

**Fact B.2.** *If $f$ is $\alpha$-strongly convex, then for any $x, y \in \mathbb{R}^d$ we have*

$$f(y) \geq f(x) + \nabla f(x)^\top (y - x) + \frac{\alpha}{2} \|y - x\|^2 .$$

The following fact gives a sufficient condition for $f$ to be $\alpha$-strongly convex.

**Fact B.3.** *If $f(x) - g(x)$ is convex and $g$ is $\alpha$-strongly convex, then $f$ is also $\alpha$-strongly convex.*

*Proof.* Define

$$h(x) \stackrel{\text{def}}{=} f(x) - g(x) = f(x) - (g(x) - \frac{\alpha}{2} \|x\|^2) - \frac{\alpha}{2} \|x\|^2 .$$

Then we have $f(x) - \frac{\alpha}{2} \|x\|^2 = h(x) + (g(x) - \frac{\alpha}{2} \|x\|^2)$. Since both functions on the RHS are convex, so is $f(x) - \frac{\alpha}{2} \|x\|^2$. □

Along any fixed direction, the directional derivative of a convex function is monotonically increasing:

**Fact B.4.** *If $f$ is a convex function, then for any $x, y \in \mathbb{R}^d$*

$$\nabla f(x)^\top (y - x) \leq \nabla f(y)^\top (y - x).$$

As we will use the idea of gradient descent in our algorithms, we will need the following fact that says that for $\alpha$-strongly convex and $\beta$-smooth functions, the negative gradient at a point has a large inner product against the direction to the minimizer $x^*$.

**Fact B.5.** *If $f$ is $\alpha$-strongly convex and $\beta$-smooth, then we have at any point $x \in \mathbb{R}^d$ that*

$$\nabla f(x)^\top (x - x^*) \geq \frac{\alpha}{2} \|x - x^*\|^2 , \tag{5}$$

$$\nabla f(x)^\top (x - x^*) \geq \frac{1}{\beta} \|\nabla f(x)\|^2 . \tag{6}$$

*Proof.* Using $\alpha$-strong convexity of $f$ we have

$$f(x) + \nabla f(x)^\top (x^* - x) + \frac{\alpha}{2} \|x^* - x\|^2 \leq f(x^*),$$

which by rearranging gives

$$\frac{\alpha}{2} \|x^* - x\|^2 \leq \nabla f(x)^\top (x - x^*) + f(x^*) - f(x) \leq \nabla f(x)^\top (x - x^*),$$

where the last inequality holds as $x^*$ is the minimizer of $f$. This proves the first claim.

To prove the second claim, we note that by convexity we have

$$f(x) + \nabla f(x)^\top (x^* - x) \leq f(x^*),$$

which in turn implies

$$\nabla f(x)^\top (x - x^*) \geq f(x) - f(x^*).$$

We further lower bound the difference between $f(x)$ and $f(x^*)$:

$$f(x) - f(x^*) \geq f(x) - f(x - \frac{1}{\beta} \nabla f(x)) \qquad (x^* \text{ is the minimizer})$$

$$\geq -\nabla f(x)^\top \left( -\frac{1}{\beta} \nabla f(x) \right) - \frac{\beta}{2} \left\| \frac{1}{\beta} \nabla f(x) \right\|^2 \qquad (\beta\text{-smoothness and Fact B.1})$$

$$= \frac{1}{2\beta} \|\nabla f(x)\|^2.$$

Combining this with the previous inequality finishes the proof. $\qquad\square$

**Lemma 2.6.** *Let $x^*, \bar{x}_r$ be the minimizers of $f$ and $\bar{f}_r$ respectively. Then $\|x^* - \bar{x}_r\| \leq 2r\sqrt{\beta/\alpha}$.*

*Proof.* As $\mathcal{B}(\bar{x}_r, r)$ is a convex set, there is a unique point $z \in \mathcal{B}(\bar{x}_r, r)$ which has the minimum distance from $x^*$. By $\alpha$-strong convexity of $f$, $f(z) - f(x^*) \geq \alpha \frac{\|z - x^*\|^2}{2}$. So, for any other point $x \in \mathcal{B}(\bar{x}_r, r)$, $f(x) - f(x^*) \geq \alpha \frac{\|z - x^*\|^2}{2}$. This concludes that $\bar{f}_r(\bar{x}_r) \geq \alpha \frac{\|z - x^*\|^2}{2}$.

On the other hand, by $\beta$-smoothness of $f$, for ever point $x \in \mathcal{B}(x^*, r)$ we have $f(x) \leq \beta r^2/2$. This concludes that $\bar{f}_r(x^*) \leq \beta r^2 / 2$.

But $\bar{x}_r$ is the minimizer of $\bar{f}_r$, so we have $\alpha \frac{\|z - x^*\|^2}{2} \leq \beta r^2 / 2$. Thus,

$$\|z - x^*\| \leq \sqrt{\beta/\alpha} r \tag{7}$$

By triangle inequality, $\|z - x^*\| + r \geq \|z - x^*\| + \|z - \bar{x}_r\| \geq \|x^* - \bar{x}_r\|$. The triangle inequality in conjunction of (7) implies Lemma 2.6. $\qquad\square$

## C   Missing proofs from Section 3

**Lemma 3.5.** *Given a set of (possibly infinitely many) tuples $\{(x_i, g_i, f_i)\}_{i \in I}$ where $x_i, g_i \in \mathbb{R}^d$, $f_i \in \mathbb{R}$ and $0 \leq \alpha < \beta \leq +\infty$. The following two statements are equivalent:*

*1. $\{(x_i, g_i, f_i)\}_{i \in I}$ is $(\alpha, \beta)$-interpolable.*

*2. $\left\{ \left( \frac{\beta x_i}{\beta - \alpha} - \frac{g_i}{\beta - \alpha}, g_i - \alpha x_i, \frac{\alpha x_i^\top g_i}{\beta - \alpha} + f_i - \frac{\beta \alpha \|x_i\|^2}{2(\beta - \alpha)} - \frac{\|g_i\|^2}{2(\beta - \alpha)} \right) \right\}_{i \in I}$ is $(0, \infty)$-interpolable.*

*Proof.* The lemma follows from the equivalence of the following propositions:

(a) $\{(x_i, g_i, f_i)\}_{i \in I}$ is $(\alpha, \beta)$-interpolable.

(b) $\left\{ \left( x_i, g_i - \alpha x_i, f_i - \frac{\alpha}{2} \|x_i\|^2 \right) \right\}_{i \in I}$ is $(0, \beta - \alpha)$-interpolable.

(c) $\left\{ \left( g_i - \alpha x_i, x_i, x_i^\top g_i - f_i - \frac{\alpha}{2} \|x_i\|^2 \right) \right\}_{i \in I}$ is $(1/(\beta - \alpha), \infty)$-interpolable.

(d) $\left\{ \left( g_i - \alpha x_i, \frac{\beta x_i}{\beta - \alpha} - \frac{g_i}{\beta - \alpha}, \frac{\beta x_i^\top g_i}{\beta - \alpha} - f_i - \frac{\beta \alpha \|x_i\|^2}{2(\beta - \alpha)} - \frac{\|g_i\|^2}{2(\beta - \alpha)} \right) \right\}_{i \in I}$ is $(0, \infty)$-interpolable.

(e) $\left\{ \left( \frac{\beta x_i}{\beta - \alpha} - \frac{g_i}{\beta - \alpha}, g_i - \alpha x_i, \frac{\alpha x_i^\top g_i}{\beta - \alpha} + f_i - \frac{\beta \alpha \|x_i\|^2}{2(\beta - \alpha)} - \frac{\|g_i\|^2}{2(\beta - \alpha)} \right) \right\}_{i \in I}$ is $(0, \infty)$-interpolable.

Here, (a)$\Leftrightarrow$(b) and (c)$\Leftrightarrow$(d) follow from Lemma 3.3, while (b)$\Leftrightarrow$(c) and (d)$\Leftrightarrow$(e) follow from Lemma 3.4. $\qquad\square$

**Proof of Lemma 3.6.** We start by presenting some useful propositions. [23] shows that if a finite set of tuples satisfies the following interpolability condition, then it is $(\alpha, \beta)$-interpolable. Although this condition does not necessarily imply the same interpolability of an infinite set of tuples, it will be useful in our analysis later. Therefore we state such a condition below and call it *finite $(\alpha, \beta)$-interpolability condition*.

**Definition C.1** (Finite $(\alpha, \beta)$-interpolability condition). Given a set of (possibly infinitely many) tuples denoted by $\{(x_i, g_i, f_i)\}_{i \in I}$ where each $x_i, g_i \in \mathbb{R}^d$, $f_i \in \mathbb{R}$ and $0 \le \alpha < \beta \le +\infty$. We say this set satisfies the finite $(\alpha, \beta)$-interpolability condition if for all pairs $i, j \in I$, we have

$$f_i - f_j - g_j^\top (x_i - x_j) \ge$$
$$\frac{1}{2(1 - \frac{\alpha}{\beta})} \left( \frac{1}{\beta} \|g_i - g_j\|^2 + \alpha \|x_i - x_j\|^2 - \frac{2\alpha}{\beta} (g_j - g_i)^\top (x_j - x_i) \right). \quad (8)$$

We give in the following fact an equivalent condition to that in Definition C.1. This fact can be verified by straightforward calculations.

**Fact C.2.** *Given a set of (possibly infinitely many) tuples $\{(x_i, g_i, f_i)\}_{i \in I}$ where each $x_i, g_i \in \mathbb{R}^d$, $f_i \in \mathbb{R}$ and $0 \le \alpha < \beta \le +\infty$. Let*

$$\hat{x}_i \overset{\text{def}}{=} \frac{\beta x_i}{\beta - \alpha} - \frac{g_i}{\beta - \alpha},$$
$$\hat{g}_i \overset{\text{def}}{=} g_i - \alpha x_i,$$
$$\hat{f}_i \overset{\text{def}}{=} \frac{\alpha x_i^\top g_i}{\beta - \alpha} + f_i - \frac{\beta \alpha \|x_i\|^2}{2(\beta - \alpha)} - \frac{\|g_i\|^2}{2(\beta - \alpha)}.$$

*Then the interpolability condition in Definition C.1 is equivalent to that for any $i, j \in I$*

$$\hat{f}_i \ge \hat{f}_j + \hat{g}_j^\top (\hat{x}_i - \hat{x}_j).$$

We will use the interpolation results from Section 3.1 to prove Lemma 3.6. Before that, let us define the points that we want to interpolate. In particular, consider the following set of tuples. For each $x \in X_{=1}$, there is a tuple $(x, \nabla f(x), f(x))$, where we recall that we have defined $f(x) \overset{\text{def}}{=} \|x\|^2$ in Lemma 3.6. In addition, there is a tuple $(\frac{1}{2} e_1, 0, \tilde{f}(\frac{1}{2} e_1))$, where we let $\tilde{f}(\frac{1}{2} e_1)$ be any number in $[\frac{5}{31}, \frac{12}{31}]$. Let $\{(x_i, g_i, f_i)\}_{i \in I}$ denote this set of tuples.

**Lemma C.3.** *Let $\alpha = 1/2$ and $\beta = 16$. Then the set $\{(x_i, g_i, f_i)\}_{i \in I}$ satisfies the finite $(\alpha, \beta)$-interpolability condition in Definition C.1.*

*Proof.* First, we already know that the function $f = \|x\|^2$ interpolates $(x_i, \nabla f(x_i), f(x_i))$'s for which $\|x_i\| = 1$. By the convexity of $f$, Statement (e) of Lemma 3.5, and Fact C.2, the condition (8) in Definition C.1 is trivially satisfied for all pairs $(x_i, \nabla f(x_i), f(x_i)), (x_j, \nabla f(x_j), f(x_j))$ for which $x_i, x_j \in X_{=1}$. Then we first consider the pairs $(x_i, \nabla f(x_i), f(x_i)), (\frac{1}{2} e_1, 0, \tilde{f}(\frac{1}{2} e_1))$ where $x_i \in X_{=1}$. In this case the LHS of (8) is

$$f(x_i) - \tilde{f}\left(\frac{1}{2} e_1\right) - 0^\top \left(x_i - \frac{1}{2} e_1\right) = 1 - \tilde{f}\left(\frac{1}{2} e_1\right) \ge \frac{19}{31},$$

where the last inequality follows from $\tilde{f}(\frac{1}{2}e_1) \leq \frac{12}{31}$. The RHS of (8) is

$$\frac{16}{31}\left(\frac{1}{16}\|2x_i - 0\|^2 + \frac{1}{2}\left\|x_i - \frac{1}{2}e_1\right\|^2 - \frac{1}{16}(0 - 2x_i)^\top(\frac{1}{2}e_1 - x_i)\right)$$

$$=\frac{16}{31}\left(\left(\frac{1}{4} + \frac{1}{2} - \frac{1}{8}\right)\|x_i\|^2 + \left(-\frac{1}{2} + \frac{1}{16}\right)e_1^\top x_i + \frac{1}{8}\|e_1\|^2\right) \qquad \text{(rearranging)}$$

$$=\frac{16}{31}\left(\frac{5}{8} - \frac{7}{16}e_1^\top x_i + \frac{1}{8}\right) \qquad (\|x_i\| = \|e_1\| = 1) \tag{9}$$

$$\leq\frac{16}{31}\left(\frac{5}{8} + \frac{7}{16} + \frac{1}{8}\right) = \frac{16}{31}\cdot\frac{19}{16} = \frac{19}{31} \qquad (e_1^\top x_i \geq -1).$$

Thus we have LHS $\geq$ RHS.

We then consider the pairs $(\frac{1}{2}e_1, 0, \tilde{f}(\frac{1}{2}e_1)), (x_j, \nabla f(x_j), f(x_j))$ where $x_j \in X_{=1}$. In this case the LHS of (8) is

$$\tilde{f}\left(\frac{1}{2}e_1\right) - f(x_j) - 2x_j^\top\left(\frac{1}{2}e_1 - x_j\right) = \tilde{f}\left(\frac{1}{2}e_1\right) - 1 - e_1^\top x_j + 2\|x_j\|^2 \geq \frac{36}{31} - e_1^\top x_j,$$

where the last inequality follows from $\|x_j\| = 1$ and $\tilde{f}(\frac{1}{2}e_1) \geq \frac{5}{31}$. Note that the RHS of (8) is symmetric with respect to $i, j$. Therefore using (9) above we have

$$\text{LHS of (8)} - \text{RHS of (8)} \geq \frac{36}{31} - e_1^\top x_j - \frac{16}{31}\left(\frac{5}{8} - \frac{7}{16}e_1^\top x_j + \frac{1}{8}\right)$$

$$= \frac{24}{31} - \frac{24}{31}e_1^\top x_j \qquad \text{(rearranging)}$$

$$\geq 0 \qquad (e_1^\top x_i \leq 1).$$

Thus again we have LHS $\geq$ RHS. This completes the proof of the lemma. $\qquad \square$

We can now prove Lemma 3.6 using the propositions above.

**Lemma 3.6.** *Let $f(x) \overset{\text{def}}{=} \|x\|^2$ which is 2-strongly convex and 2-smooth. There exists a $\frac{1}{2}$-strongly convex, 16-smooth function $\tilde{f}$ such that*

1. *$\tilde{f}$'s minimizer is $\frac{1}{2}e_1$.*

2. *For all $x \in X_{=1}$ we have $\tilde{f}(x) = f(x)$ and $\nabla\tilde{f}(x) = \nabla f(x)$.*

*Proof.* Let $\hat{x}_i, \hat{g}_i, \hat{f}_i$ be defined as in Fact C.2 for $\alpha = 1/2$ and $\beta = 16$. Then by Lemma C.3 and Fact C.2, we have for all $i, j \in I$

$$\hat{f}_i \geq \hat{f}_j + \hat{g}_j^\top(\hat{x}_i - \hat{x}_j). \tag{10}$$

We will show that $\left\{(\hat{x}_i, \hat{g}_i, \hat{f}_i)\right\}_{i\in I}$ is $(0, \infty)$-interpolable by constructing a proper and closed convex function $\varphi$ interpolating them. This coupled with Lemma 3.5 implies that $\{(x_i, g_i, f_i)\}_{i\in I}$ is $(1/2, 16)$-interpolable. Specifically, we define $\varphi$ by

$$\varphi(x) \overset{\text{def}}{=} \sup\left\{\hat{f}_j + \hat{g}_j^\top(x - \hat{x}_j) : j \in I\right\}.$$

Immediately from the definition of $\varphi$ and (10) we have that $\varphi$ is convex (as it is the pointwise supremum of affine functions) and $\varphi(\hat{x}_i) = \hat{f}_i$ and $\hat{g}_i \in \partial\varphi(\hat{x}_i)$ for all $i \in I$. We then show that $\varphi$ is proper and closed.

Since for the quadratic function $f(x) = \|x\|^2$ we have $\nabla f(x) = 2x$, we have for all $x_i \in X_{=1}$ that $\hat{x}_i = \frac{\beta-2}{\beta-\alpha}x_i$, $\hat{g}_i = (2-\alpha)x_i$, and $\hat{f}_i$'s are all equal. Then for any $x$, if we let $j_1(x)$ be such that $x_{j_1(x)} = x/\|x\|$, we have

$$\sup\left\{\hat{f}_j + \hat{g}_j^\top(x - \hat{x}_j) : x_j \in X_{=1}\right\} = \hat{f}_{j_1(x)} + \hat{g}_{j_1(x)}^\top(x - \hat{x}_{j_1(x)}) = A\|x\| + B$$

for constants $A, B \in \mathbb{R}$. Let $j'$ be such that $x_{j'} = \frac{1}{2}e_1$. Then as a result of the above equation, we can write $\varphi$ as

$$\varphi(x) = \max\left\{\hat{f}_{j'}, A\|x\| + B\right\}.$$

Since $\varphi$ is the pointwise max of two proper and closed convex functions, it is proper and closed as well. $\qquad\square$

**Lemma 3.7.** *Let* $f(x) \overset{\text{def}}{=} \|x\|^2$ *which is 2-strongly convex and 2-smooth. Define* $\hat{f}$ *such that* $\hat{f}(x) = \tilde{f}(x)$ *if* $\|x\| \leq 1$ *and* $\hat{f}(x) = f(x)$ *otherwise* ($\|x\| > 1$). *Then we have*

1. $\hat{f}$ *is* $\frac{1}{2}$*-strongly convex and 16-smooth.*

2. $\hat{f}$*'s minimizer is* $\frac{1}{2}e_1$.

3. *For all* $x \in X_{\geq 1}$ *we have* $\hat{f}(x) = f(x)$ *and* $\nabla\hat{f}(x) = \nabla f(x)$.

*Proof.* Since by Lemma 3.6 both the function values and gradients of $\tilde{f}$ agree with those of $f$ on $X_{=1}$, $\hat{f}$ is differentiable and its minimal value is obtained at $\frac{1}{2}e_1$ where its gradient is 0. Also $\hat{f}$'s function values and gradients agree with those of $f$ on $X_{\geq 1}$ by its definition. We then show that $\hat{f}$ is $1/2$-strongly convex and 16-smooth.

1. (1/2-strong convexity) Define $h, \tilde{h}, \hat{h}$ as $f(x) - \frac{1}{4}\|x\|^2$, $\tilde{f}(x) - \frac{1}{4}\|x\|^2$, and $\hat{f}(x) - \frac{1}{4}\|x\|^2$ respectively. We need to show that $\hat{h}$ is convex, i.e.

$$\hat{h}(x) + \nabla\hat{h}(x)^\top(y - x) \leq \hat{h}(y), \quad \forall x, y \in \mathbb{R}^d. \tag{11}$$

Due to the $1/2$-strong convexity of $f$ and $\tilde{f}$, (11) immediately holds when $\|x\|, \|y\|$ are both larger than 1 or both smaller than 1. When it is the case that $\|x\| < 1 < \|y\|$, let $z$ be the unique point on the line segment $xy$ with $\|z\| = 1$. Then we have

$$\begin{aligned}
\hat{h}(x) + \nabla\hat{h}(x)^\top(y - x) =& \tilde{h}(x) + \nabla\tilde{h}(x)^\top(z - x) + \nabla\tilde{h}(x)^\top(y - z) \\
\leq& \tilde{h}(z) + \nabla\tilde{h}(x)^\top(y - z) \quad (\tilde{h} \text{ convex}) \\
\leq& \tilde{h}(z) + \nabla\tilde{h}(z)^\top(y - z) \quad (\text{Fact B.4}) \\
=& h(z) + \nabla h(z)^\top(y - z) \leq h(y) = \hat{h}(y) \quad (h \text{ convex}).
\end{aligned}$$

The same reasoning shows that (11) also holds when $\|y\| < 1 < \|x\|$.

2. (16-smoothness) We need to show that

$$\left\|\nabla\hat{f}(x) - \nabla\hat{f}(y)\right\| \leq 16\|x - y\|, \quad \forall x, y \in \mathbb{R}^d. \tag{12}$$

Due to the 16-smoothness of $f$ and $\tilde{f}$, (12) once again immediately holds when $\|x\|, \|y\|$ are both larger than 1 or both smaller than 1. When it is the case that $\|x\| < 1 < \|y\|$, let $z$ be the unique point on the line segment $xy$ with $\|z\| = 1$. Then we have

$$\begin{aligned}
\left\|\nabla\hat{f}(x) - \nabla\hat{f}(y)\right\| =& \left\|\nabla\tilde{f}(x) - \nabla\tilde{f}(z) + \nabla f(z) - \nabla f(y)\right\| \\
\leq& \left\|\nabla\tilde{f}(x) - \nabla\tilde{f}(z)\right\| + \|\nabla f(z) - \nabla f(y)\| \quad (\text{triangle inequality}) \\
\leq& 16\|x - z\| + 16\|z - y\| \quad (\text{16-smoothness of } \tilde{f} \text{ and } f) \\
=& 16\|x - y\| \quad (z \text{ on line segment } xy).
\end{aligned}$$

The same reasoning shows that (12) also holds when $\|y\| < 1 < \|x\|$.

This completes the proof. $\qquad\square$

**Lemma 3.8.** *Given $\kappa \geq 1$ with $1+\log\kappa \leq d$ where $d$ is the dimension. Let $\gamma \overset{\text{def}}{=} (1/\kappa)^{\frac{1}{d-1}} \in [1/2, 1]$. Let $S_{d\times d} = \mathrm{DIAG}(\kappa, \gamma, \ldots, \gamma)$. Define $s(x) = x^\top S^{-1} x$, which is $(2/\kappa)$-strongly convex and $(2/\gamma)$-smooth. Let $X_{s\geq 1} = \{x : s(x) \geq 1\}$. Also define $\hat{s}(x) = \hat{f}(S^{-1/2}x)$. Then we have*

    *1. $\hat{s}$ is $1/(2\kappa)$-strongly convex and $(16/\gamma)$-smooth.*

    *2. $\hat{s}$'s minimizer is $\frac{\sqrt{\kappa}}{2}e_1$.*

    *3. For all $x \in X_{s\geq 1}$ we have $\hat{s}(x) = s(x)$ and $\nabla\hat{s}(x) = \nabla s(x)$.*

*Proof.* Let $f(x) \overset{\text{def}}{=} \|x\|^2$ which is 2-strongly convex and 2-smooth. Let $X_{\geq 1} \overset{\text{def}}{=} \{x : \|x\| \geq 1\}$. Let $\hat{f}$ be a $1/2$-strongly convex and 16-smooth function whose minimal value is obtained at $\frac{1}{2}e_1$ and whose values and gradients agree with $f$ on $X_{\geq 1}$ (Lemma 3.7).

We note that $s(x) = f(S^{-1/2}x)$. Also observe that

$$\nabla\hat{s}(x) = S^{-1/2}\nabla\hat{f}(S^{-1/2}x). \tag{13}$$

Thus $\hat{s}$ obtains its minimal value at point $S^{1/2}(\frac{1}{2}e_1) = \frac{\sqrt{\kappa}}{2}e_1$ where its gradient is zero. Moreover $\hat{s}$'s function values and gradients agree with those of $s$ on $\left\{x : \|S^{-1/2}x\| \geq 1\right\} = X_{s\geq 1}$.

Due to the $1/2$-strong convexity of $\hat{f}$, we know that $\hat{f}(x) - \frac{1}{4}\|x\|^2$ is convex. Since convexity is preserved under linear transformation over the domain, the function $\hat{s}(x) - \frac{1}{4}\|S^{-1/2}x\|^2$ is also convex. As $\frac{1}{4}\|S^{-1/2}x\|^2 = \frac{1}{4}x^\top S^{-1}x$ is itself $1/(2\kappa)$-strongly convex, so is $\hat{s}$ (Fact B.3).

To show the $(16/\gamma)$-smoothness of $\hat{s}$, we note that by (13) for all $x, y \in \mathbb{R}^d$,

$$
\begin{aligned}
\|\nabla\hat{s}(x) - \nabla\hat{s}(y)\| &= \left\|S^{-1/2}\left(\nabla\hat{f}(S^{-1/2}x) - \nabla\hat{f}(S^{-1/2}y)\right)\right\| \\
&\leq \gamma^{-1/2}\left\|\nabla\hat{f}(S^{-1/2}x) - \nabla\hat{f}(S^{-1/2}y)\right\| \\
&\leq \gamma^{-1/2}\cdot 16\left\|S^{-1/2}(x-y)\right\| \qquad \text{(16-smoothness of } \hat{f}) \\
&\leq (16/\gamma)\|x-y\|.
\end{aligned}
$$

This finishes the proof of the lemma. $\qquad\square$

**Theorem 3.1.** *Given $0 < \alpha \leq \beta$ with $1 + \log\frac{\beta}{\alpha} \leq d$ where $d$ is the dimension, and a $K > 0$, there exist two $\alpha$-strongly convex, $\beta$-smooth functions whose values differ only in an ellipsoid of volume equal to a radius-$K$ ball, but whose minimizers are $\Omega(\sqrt{\frac{\beta}{\alpha}}K)$-far from each other.*

*Proof.* Set $\kappa = \frac{\beta}{64\alpha}$ and let $\gamma$, $S$, $s$, and $\hat{s}$ be as defined in Lemma 3.8. Then we know that $s$ and $\hat{s}$ are both $\frac{32\alpha}{\beta}$-strongly convex and 32-smooth. Now we define

$$f(x) \overset{\text{def}}{=} \frac{\beta}{32}\cdot K^2 \cdot s\left(\frac{1}{K}x\right) \qquad \text{and} \qquad \hat{f}(x) \overset{\text{def}}{=} \frac{\beta}{32}\cdot K^2 \cdot \hat{s}\left(\frac{1}{K}x\right).$$

From the definitions $f$ and $\hat{f}$ are both $\alpha$-strongly convex and $\beta$-smooth. Moreover, note that the function values of $s$ and $\hat{s}$ only differ in the ellipsoid $x^\top S^{-1}x \leq 1$, whose volume equals that of a unit ball since the determinant of $S$ is 1 by the definition in Lemma 3.8. As a result, the function values of $f$ and $\hat{f}$ only differ in an ellipsoid of volume equal to a radius-$K$ ball. Finally, note that $s$ and $\hat{s}$ obtain their minimal values at points 0 and $\frac{\sqrt{\kappa}}{2}e_1 = \frac{1}{16}\sqrt{\frac{\beta}{\alpha}}e_1$ respectively. Thus $f$ and $\hat{f}$ obtain their minimal values at 0 and $\frac{1}{16}\sqrt{\frac{\beta}{\alpha}}Ke_1$ respectively, which are $\frac{1}{16}\sqrt{\frac{\beta}{\alpha}}K$ apart. $\qquad\square$

## D  Missing proofs from Section 4

**Lemma 4.1.** *Fix $d > 0$ and $\beta > 0$. There exists a function $\texttt{err}(\tau)$ satisfying $\lim_{\tau \to 0^+} \texttt{err}(\tau) = 0$ such that the following holds. Fix any $x \in \mathbb{R}^d$ and $\tau > 0$ such that the radius-$\tau$ ball centered at $x$ is mostly uncorrupted:*

$$\Pr_{y \sim \mathcal{U}(x, \tau)} \left[ f(y) \neq \hat{f}(y) \right] \leq \frac{1}{100}. \tag{2}$$

*Then we have that with probability $1 - 2^{-3d}$, the vector $g$ returned by GRADIENTCOMP satisfies*

$$\|g - \nabla f(x)\| \leq \texttt{err}(\tau). \tag{3}$$

*The number of queries made by GRADIENTCOMP is $O(d)$.*

*Proof.* We say a pair $a_j, b_j$ is uncorrupted if $\hat{f}(a_j) = f(a_j)$ and $\hat{f}(b_j) = f(b_j)$. Then each pair is uncorrupted with probability $\geq 0.99^2 > 0.98$. Using a Chernoff bound we have that with probability $1 - 2^{-3d}$ there are at least $800d$ uncorrupted pairs. In this case, the vector $\nabla f(x)$ certainly satisfies the condition (1) for the $800d$ uncorrupted pairs by the $\beta$-smoothness of $f$. Moreover, for any vector $g$ satisfying (1) for at least $800d$ pairs, it must satisfy (1) for at least $600d$ uncorrupted pairs (as the number of corrupted pairs is at most $200d$). This means that when $\tau \to 0^+$, $g$ has to be arbitrarily close to the solution to the system of linear equations

$$g^\top (b_j - a_j) = \hat{f}(b_j) - \hat{f}(a_j)$$

for all $600d$ uncorrupted $j$'s. The solution to the above linear system itself becomes arbitrarily close to $\nabla f(x)$ as $\tau \to 0^+$. Therefore the existence of a desired $\texttt{err}(\tau)$ follows. $\qquad \square$

**Lemma 4.3.** *Let $d \geq 2$. The $\hat{g}$ computed at Line 6 of GDSTAGEI satisfies with probability $1 - \frac{\delta}{T}$ that $\|\hat{g} - \nabla f(x_t)\| \leq 200\beta K$.*

*Proof.* Let $\tau$ be a sufficiently small parameter used in GRADIENTCOMP such that $\texttt{err}(\tau) \leq \upsilon$ for $\upsilon = \beta K / 2$. We say a point $y_i$ is good if the fraction of corruption in the ball $\mathcal{B}(y_i, \tau)$ is at most $1/100$, i.e.

$$\Pr_{y \sim \mathcal{U}(y_i, \tau)} \left[ f(y) \neq \hat{f}(y) \right] \leq \frac{1}{100}.$$

As $d \geq 2$, the volume of $\mathcal{B}(x_t, 99K)$ is at least $9801$ times larger than a ball of radius $K$. Therefore each $y_i$ is good with probability $0.98$ by Markov's inequality. If $y_i$ is good, then by Lemma 4.1 we have $\|g_i - \nabla f(y_i)\| \leq \upsilon$ with probability at least $1 - 2^{-3d} \geq 1 - 2^{-6} > 0.98$. Multiplying these two probabilities together we have that $\|g_i - \nabla f(y_i)\| \leq \upsilon$ with probability at least $0.96$. Then using a Chernoff bound, we have with probability at least $1 - 2^{-s/60} \geq 1 - \frac{\delta}{T}$ that $\|g_i - \nabla f(y_i)\| \leq \upsilon$ holds for $(2s)/3$ of the $i$'s. When this happens, the vector $\nabla f(x_t)$ certainly satisfies the requirement of $\hat{g}$ by the $\beta$-smoothness of $f$. Moreover, any $\hat{g}$ satisfying the requirement must be within distance $99.5\beta K$ of $(s/3)$ of the $\nabla f(y_i)$'s, which themselves are within distance $99\beta K$ of $\nabla f(x_t)$. This implies that $\|\hat{g} - \nabla f(x_t)\| \leq 200\beta K$ holds with probability $1 - \frac{\delta}{T}$. $\qquad \square$

We now prove Theorem 4.2.

**Theorem 4.2.** *Let $d \geq 2$. Given an initial point $x_0$ with $\|x_0 - x^*\| \leq R_0$ and a $\delta \in (0, 1)$, the algorithm GDSTAGEI returns a point $\hat{x}$ with $\|\hat{x} - x^*\| \leq 10000(\beta/\alpha)K$ with probability $1 - \delta$, where $x^*$ is the minimizer of $f$. The number of queries made by GDSTAGEI is $\tilde{O}(d(\beta/\alpha) \log \frac{R_0}{(\beta/\alpha)K} \log(1/\delta))$.*

*Proof.* The bound on the number of queries follows straightforwardly. To show the guarantee on $\hat{x}$ returned by the algorithm, first note that by Lemma 4.3 and a union bound, with probability $1 - \delta$, at all steps the vector $\hat{g}$ satisfies $\|\hat{g} - \nabla f(x_t)\| \leq 200\beta K$. We then only need to show that when this

holds, the algorithm finds an $\hat{x}$ with $\|\hat{x} - x^*\| \leq 10000(\beta/\alpha)K$. To this end, we consider how much progress we can make in one descent step. Namely, we have

$$\|x_{t+1} - x^*\|^2 = \left\| x_t - \frac{1}{2\beta}\hat{g} - x^* \right\|^2 = \|x_t - x^*\|^2 - \frac{1}{\beta}\hat{g}^\top(x_t - x^*) + \frac{1}{4\beta^2}\|\hat{g}\|^2$$

$$= \left( \|x_t - x^*\|^2 - \frac{1}{2\beta}\hat{g}^\top(x_t - x^*) \right) - \frac{1}{2\beta}\hat{g}^\top(x_t - x^*) + \frac{1}{4\beta^2}\|\hat{g}\|^2. \qquad (14)$$

We then bound the three terms in the above equation separately:

$$\|x_t - x^*\|^2 - \frac{1}{2\beta}\hat{g}^\top(x_t - x^*)$$

$$\leq \|x_t - x^*\|^2 - \frac{1}{2\beta}\nabla f(x_t)^\top(x_t - x^*) + \frac{1}{2\beta}\|\hat{g} - \nabla f(x_t)\|\,\|x_t - x^*\| \qquad \text{(Cauchy-Schwarz)}$$

$$\leq \left( 1 - \frac{\alpha}{4\beta} \right) \|x_t - x^*\|^2 + 100K\,\|x_t - x^*\| \qquad \text{(Fact B.5, } \|\hat{g} - \nabla f(x_t)\| \leq 200\beta K\text{),}$$

also

$$-\frac{1}{2\beta}\hat{g}^\top(x_t - x^*) = -\frac{1}{2\beta}\nabla f(x_t)^\top(x_t - x^*) + \frac{1}{2\beta}(\nabla f(x_t) - \hat{g})^\top(x_t - x^*)$$

$$\leq -\frac{1}{2\beta}\nabla f(x_t)^\top(x_t - x^*) + 100K\,\|x_t - x^*\| \qquad \text{(Cauchy-Schwarz)}$$

$$\leq -\frac{1}{2\beta^2}\|\nabla f(x_t)\|^2 + 100K\,\|x_t - x^*\| \qquad \text{(Fact B.5),}$$

and finally

$$\frac{1}{4\beta^2}\|\hat{g}\|^2$$

$$= \frac{1}{4\beta^2}\left( \|\nabla f(x_t)\|^2 + 2(\hat{g} - \nabla f(x_t))^\top\nabla f(x_t) + \|\hat{g} - \nabla f(x_t)\|^2 \right)$$

$$\leq \frac{1}{4\beta^2}\left( \|\nabla f(x_t)\|^2 + 2(200\beta K)\|\nabla f(x_t)\| + (200\beta K)^2 \right) \qquad \text{(Cauchy-Schwarz)}$$

$$\leq \frac{1}{4\beta^2}\|\nabla f(x_t)\|^2 + 100K\,\|x_t - x^*\| + 10000K^2 \qquad \text{(}\beta\text{-smoothness).}$$

Plugging these three upper bounds into (14) and noting that $\left( -\frac{1}{2\beta^2} + \frac{1}{4\beta^2} \right)\|\nabla f(x_t)\|^2 \leq 0$:

$$\|x_{t+1} - x^*\|^2 \leq \left( 1 - \frac{\alpha}{4\beta} \right) \|x_t - x^*\|^2 + 300K\,\|x_t - x^*\| + 10000K^2.$$

If $\|x_t - x^*\| \geq 3000(\beta/\alpha)K$, we have

$$\|x_{t+1} - x^*\|^2 \leq \left( 1 - \frac{\alpha}{8\beta} \right) \|x_t - x^*\|^2.$$

Therefore after $\log_{(1-\alpha/(8\beta))^{-1}} \frac{R_0}{(\beta/\alpha)K} \leq T$ steps, $x_t$ will stay within distance $10000(\beta/\alpha)K$ of the minimizer $x^*$ of $f$. This completes the proof of the theorem. □

**Claim 4.6.** *A vector $\hat{g}$ satisfying the condition at Line 6 of* GDSTAGEI, *if it exists, can be found in nearly-linear time in $s$, at the cost of an extra constant factor in the radius of the ball.*

*Proof of Claim 4.6.* We show that if there exists a $\hat{g}$ such that the ball $\mathcal{B}(\hat{g}, 100\beta K)$ traps at least $\frac{2s}{3}$ of the $g_i$'s, then there is an algorithm that runs in $O(s\log(1/\delta))$ time and finds an index $j$ such that the ball $\mathcal{B}(g_j, 200\beta K)$ traps at least $\frac{2s}{3}$ of the $g_i$'s with probability $1 - \delta$, for any $\delta > 0$.

The algorithm works by repeating the following for $100\log(1/\delta)$ times: pick a random index $j$ and check if the ball $\mathcal{B}(g_j, 200\beta K)$ traps at least $\frac{2s}{3}$ of the $g_i$'s. We then output any $g_j$ that passes the test. Note that if a desired $\hat{g}$ exists, then the points in the ball $\mathcal{B}(\hat{g}, 100\beta K)$, which are $\frac{2}{3}$ fraction of the total, can all pass the test. Thus each test succeeds with probability at least $2/3$. Consequently there is at least one successful test out of $100\log(1/\delta)$ with probability at least $1 - \delta$. □

# E  Missing proofs from Section 5

**Theorem E.1** (Vector Bernstein Inequality, Lemma 18 of [16])**.** *Let $z_1, z_2, \ldots, z_s \in \mathbb{R}^d$ be independent random vectors. Suppose each random vector satisfies*

$$\mathbb{E}\left[z_i\right] = 0 \qquad and \qquad \|z_i\| \leq R \qquad and \qquad \mathbb{E}\left[\|z_i\|^2\right] \leq \sigma^2$$

*for some $R > 0$ and $\sigma^2 > 0$. Then, for all $0 < \epsilon < \sigma^2/R$,*

$$\Pr\left[\left\|\frac{1}{s}\sum_{i=1}^{s} z_i\right\| \geq \epsilon\right] \leq \exp\left(-s \cdot \frac{\epsilon^2}{8\sigma^2} + \frac{1}{4}\right).$$

**Lemma 5.2.** *Let $d \geq 100\log(\beta/\alpha + 1)$. The vector $\bar{g}$ computed at Line 5 of $\mathrm{GDSTAGEII}$ satisfies the following with probability at least $1 - 2^{-d/8}/T$: $\left\|\bar{g} - \nabla \bar{f}_{2K}(x_t)\right\| \leq 16\sqrt{\alpha\beta}K$.*

*Proof.* Let $\tau$ be a sufficiently small parameter used in $\mathrm{GRADIENTCOMP}$ such that $\mathtt{err}(\tau) \leq \upsilon$ for some $\upsilon > 0$ to be specified later. We say a point $y_i$ is good if the fraction of corruption in the ball $\mathcal{B}(y_i, \tau)$ is at most $1/100$, i.e.

$$\Pr_{y \sim \mathcal{U}(y_i, \tau)}\left[f(y) \neq \hat{f}(y)\right] \leq \frac{1}{100}.$$

The volume of $\mathcal{B}(x_t, 2K)$ is at least $2^d$ times larger than a ball of radius $K$. Therefore each $y_i$ is good with probability $1 - 100 \cdot 2^{-d}$ by Markov's inequality. If $y_i$ is good, then by Lemma 4.1 we have $\|g_i - \nabla f(y_i)\| \leq \upsilon$ with probability at least $1 - 2^{-3d}$. Multiplying these two probabilities together we have that $\|g_i - \nabla f(y_i)\| \leq \upsilon$ with probability at least $1 - 200 \cdot 2^{-d}$. Using the fact that $d \geq 100\log(\beta/\alpha + 1)$ and a union bound, we have that $\|g_i - \nabla f(y_i)\| \leq \upsilon$ holds for all $i$'s with probability at least $1 - 2^{-d/2}$. Furthermore, since $T \leq 2^{d/4}$, this probability is at least $1 - 2^{-d/4}/T$.

We then consider the probability that $\left\|\bar{g} - \nabla \bar{f}_{2K}(x_t)\right\| \leq 9\sqrt{\alpha\beta}K$ holds if the ball $\mathcal{B}(x_t, 2K)$ was not corrupted at all (i.e. $\hat{f}(y) = f(y), \forall y \in \mathcal{B}(x_t, 2K)$). Suppose for a moment we have exactly $g_i = \nabla f(y_i)$ for all $i$'s (we will incorporate these errors later). We know that for each $i$, $\nabla f(y_i) = \nabla \bar{f}_{2K}(x_t)$ in expectation. Moreover, since $f$ is $\beta$-smooth, we have $\left\|\nabla f(y_i) - \nabla \bar{f}_{2K}(x_t)\right\| \leq 4\beta K$ for all $i$. We now invoke Theorem E.1 to prove concentration. Let $z_i \stackrel{\text{def}}{=} \nabla f(y_i) - \nabla \bar{f}_{2K}(x_t)$. Then we have $\mathbb{E}\left[z_i\right] = 0$, $\|z_i\| \leq 4\beta K$, and $\mathbb{E}\left[\|z_i\|^2\right] \leq 16\beta^2 K^2$. Using Theorem E.1 with $\epsilon = 9\sqrt{\alpha\beta}K$ gives

$$\begin{aligned}
\Pr\left[\left\|\bar{g} - \nabla \bar{f}_{2K}(x_t)\right\| \geq 16\sqrt{\alpha\beta}K\right] &\leq \exp\left(-s(9^2\alpha\beta K^2)/(8 \cdot 16\beta^2 K^2) + 1/4\right) \\
&\leq \exp\left(-200\log(dT) + 1/4\right) \\
&\leq 2^{-100d}/T,
\end{aligned}$$

where the second line follows from that $s \geq 400\frac{\beta}{\alpha}\log(dT)$. Now by setting $\upsilon = \sqrt{\alpha\beta}K$, and combining the above inequality with our previous argument that $\|g_i - \nabla f(y_i)\| \leq \upsilon$ holds for all $i$'s with probability at least $1 - 2^{-d/4}/T$, we have $\Pr\left[\left\|\bar{g} - \nabla \bar{f}_{2K}(x_t)\right\| \geq 16\sqrt{\alpha\beta}K\right] \leq 1 - 2^{-d/8}/T$ as desired. $\qquad\square$

We now prove Theorem 5.1.

**Theorem 5.1.** *Suppose that $d \geq 100\log(\beta/\alpha + 1)$. Then given an initial point $x_0$ that satisfies $\|x_0 - x^*\| \leq 10000(\beta/\alpha)K$, $\mathrm{GDSTAGEII}$ returns a point $\hat{x}$ with $\|\hat{x} - x^*\| \leq 1000\sqrt{\beta/\alpha}K$ with probability at least $1 - 2^{-d/8}$, where $x^*$ is the minimizer of $f$. The number of queries made by $\mathrm{GDSTAGEII}$ is $\tilde{O}(d(\beta/\alpha)^2)$. Moreover, the algorithm runs in polynomial time.*

*Proof.* The bound on the number of queries follows straightforwardly. To show the guarantee on $\hat{x}$ returned by the algorithm, first note that by Lemma 5.2 and a union bound, with probability $1 - 2^{-d/8}$, at all steps the vector $\bar{g}$ satisfies $\left\|\bar{g} - \nabla \bar{f}_{2K}(x_t)\right\| \leq 16\sqrt{\alpha\beta}K$. We then only need to show that

when this holds, the algorithm finds an $\hat{x}$ with $\|\hat{x} - x^*\| \leq 1000\sqrt{\beta/\alpha}K$. To this end, we will show that $\|\hat{x} - \bar{x}\| \leq 500\sqrt{\beta/\alpha}K$, where $\bar{x}$ is the minimizer of $\bar{f}_{2K}$. Then by Lemma 2.6 we have the desired bound on $\|\hat{x} - x^*\|$.

By Lemma 2.6 and $\|x_0 - x^*\| \leq 10000(\beta/\alpha)K$, we have $\|x_0 - \bar{x}\| \leq 20000(\beta/\alpha)K$. We then consider how much progress we can make in one descent step. Namely, we have

$$\|x_{t+1} - \bar{x}\|^2 = \left\| x_t - \frac{1}{2\beta}\bar{g} - \bar{x} \right\|^2 = \|x_t - \bar{x}\|^2 - \frac{1}{\beta}\bar{g}^\top(x_t - \bar{x}) + \frac{1}{4\beta^2}\|\bar{g}\|^2$$

$$= \left( \|x_t - \bar{x}\|^2 - \frac{1}{2\beta}\bar{g}^\top(x_t - \bar{x}) \right) - \frac{1}{2\beta}\bar{g}^\top(x_t - \bar{x}) + \frac{1}{4\beta^2}\|\bar{g}\|^2. \quad (15)$$

We then bound the three terms in the above equation separately:

$$\|x_t - \bar{x}\|^2 - \frac{1}{2\beta}\bar{g}^\top(x_t - \bar{x})$$

$$\leq \|x_t - \bar{x}\|^2 - \frac{1}{2\beta}\nabla\bar{f}_{2K}(x_t)^\top(x_t - \bar{x}) + \frac{1}{2\beta}\left\|\bar{g} - \nabla\bar{f}_{2K}(x_t)\right\| \|x_t - \bar{x}\| \quad \text{(Cauchy-Schwarz)}$$

$$\leq \left( 1 - \frac{\alpha}{4\beta} \right)\|x_t - \bar{x}\|^2 + 8\sqrt{\alpha/\beta}K\|x_t - \bar{x}\| \quad \text{(Fact B.5, } \|\bar{g} - \nabla\bar{f}_{2K}(x_t)\| \leq 16\sqrt{\alpha\beta}K\text{),}$$

also

$$-\frac{1}{2\beta}\bar{g}^\top(x_t - \bar{x}) = -\frac{1}{2\beta}\nabla\bar{f}_{2K}(x_t)^\top(x_t - \bar{x}) + \frac{1}{2\beta}(\nabla\bar{f}_{2K}(x_t) - \bar{g})^\top(x_t - \bar{x})$$

$$\leq -\frac{1}{2\beta}\nabla\bar{f}_{2K}(x_t)^\top(x_t - \bar{x}) + 8\sqrt{\alpha/\beta}K\|x_t - \bar{x}\| \quad \text{(Cauchy-Schwarz)}$$

$$\leq -\frac{1}{2\beta^2}\left\|\nabla\bar{f}_{2K}(x_t)\right\|^2 + 8\sqrt{\alpha/\beta}K\|x_t - \bar{x}\| \quad \text{(Fact B.5),}$$

and finally

$$\frac{1}{4\beta^2}\|\bar{g}\|^2$$

$$=\frac{1}{4\beta^2}\left( \left\|\nabla\bar{f}_{2K}(x_t)\right\|^2 + 2(\bar{g} - \nabla\bar{f}_{2K}(x_t))^\top\nabla\bar{f}_{2K}(x_t) + \left\|\bar{g} - \nabla\bar{f}_{2K}(x_t)\right\|^2 \right)$$

$$\leq\frac{1}{4\beta^2}\left( \left\|\nabla\bar{f}_{2K}(x_t)\right\|^2 + 2(16\sqrt{\alpha\beta}K)\left\|\nabla\bar{f}_{2K}(x_t)\right\| + (16\sqrt{\alpha\beta}K)^2 \right) \quad \text{(Cauchy-Schwarz)}$$

$$\leq\frac{1}{4\beta^2}\left\|\nabla\bar{f}_{2K}(x_t)\right\|^2 + 8\sqrt{\alpha/\beta}K\|x_t - \bar{x}\| + 64(\alpha/\beta)K^2 \quad \text{($\beta$-smoothness).}$$

Plugging these three upper bounds into (15) and noting that $\left(-\frac{1}{2\beta^2} + \frac{1}{4\beta^2}\right)\left\|\nabla\bar{f}_{2K}(x_t)\right\|^2 \leq 0$:

$$\|x_{t+1} - \bar{x}\|^2 \leq \left( 1 - \frac{\alpha}{4\beta} \right)\|x_t - \bar{x}\|^2 + 24\sqrt{\alpha/\beta}K\|x_t - \bar{x}\| + 64(\alpha/\beta)K^2.$$

If $\|x_t - \bar{x}\| \geq 200\sqrt{\beta/\alpha}K$, we have

$$\|x_{t+1} - x^*\|^2 \leq \left( 1 - \frac{\alpha}{8\beta} \right)\|x_t - x^*\|^2.$$

Therefore after $\log_{(1-\alpha/(8\beta))^{-1}} \frac{\|x_0 - \bar{x}\|}{\sqrt{\beta/\alpha}K} \leq T$ steps, $x_t$ will stay within distance $500\sqrt{\beta/\alpha}K$ of the minimizer $\bar{x}$ of $\bar{f}_{2K}$. This coupled with Lemma 2.6 completes the proof of the theorem. $\qquad\square$

We finally give a corollary stating that the success probability of the algorithm GDSTAGEII can be boosted to arbitrarily high with only a small overhead in efficiently.

**Corollary E.2** (Of Theorem 5.1). *Suppose that $d \geq 100\log(\beta/\alpha + 1)$. Then given an initial point $x_0$ that satisfies $\|x_0 - x^*\| \leq 10000(\beta/\alpha)K$, and a $\delta \in (0,1)$, we can find a point $\hat{x}$ with $\|\hat{x} - x^*\| \leq 2000\sqrt{\beta/\alpha}K$ with probability at least $1 - \delta$, where $x^*$ is the minimizer of $f$. The number of queries needed is $\tilde{O}(d(\beta/\alpha)^2\log(1/\delta))$.*

*Proof.* We run the algorithm GDSTAGEII for $100 \log(1/\delta)$ times independently, and then output a point $x$ such that the ball $\mathcal{B}(x, 1000\sqrt{\beta/\alpha})$ traps $2/3$ fraction of the points returned by GDSTAGEII's. Then by Theorem 5.1 and a Chernoff bound, it is not hard to see that $x$ has to be $1000\sqrt{\beta/\alpha}$-close to some point which itself is $1000\sqrt{\beta/\alpha}$-close to $x^*$. Finally, we note that similar to Claim 4.6 we can find such an $x$ in time nearly-linear in $\log(1/\delta)$ at the cost of an extra constant factor in the radius of the ball. $\square$