# OpenReview forum: "Approximate optimization of convex functions with outlier noise"
_NeurIPS.cc/2021/Conference — NeurIPS 2021 Poster_

### Official Review · Reviewer_E9DP · 2021-07-12

**Rating:** 6
**Confidence:** 3

**Summary:**

The paper studies the problem of convex optimization using a zeroth order oracle with adversarial noise.  In particular, the paper studies the setting where there is some convex function $f: \mathbb{R}^d \rightarrow \mathbb{R}$ and the learner may interact with an oracle that returns the value of $f$ at a given point except the oracle is corrupted on some set of measure equal to $\textsf{Vol}(B(0,K))$ (where $B(0,K)$ is the ball of radius $K$) on which the oracle may return an arbitrary value.

The paper gives an algorithm that outputs a point that is within $K \sqrt{\beta/\alpha}$ of the minimizer of $f$ (where $\beta/\alpha$ is the condition number of $f$).  They also prove a matching information-theoretic lower bound that no algorithm can guarantee to get closer than $K \sqrt{\beta/\alpha}$.  At a high level, their algorithm is based on estimating the gradient at each point using the corrupted zeroth order oracle and then running gradient-descent.

**Main Review:**

The overall contributions are ok.  The presentation is also reasonable, but occasionally handwavy, which is understandable due to space constraints.  One point that was not clear to me was the following: in the algorithm description, what does it mean for a ball centered at $\widehat{g}$ to trap another vector $g_i$?  Does it just mean that the ball contains the vector?


I have some concerns about the overall significance of the result because of limitations in the problem setup.  My concern is that the authors rely on the dimension being sufficiently high so that failure probabilities from sampling are small.  But in high dimensions, a ball of radius $K$ only represents an exponentially small fraction of the entire domain so the result is really only able to deal with an exponentially small fraction of noise.

Also, one minor comment: it seems that with $\alpha$ and $\beta$, the only thing that actually matters is the ratio $\beta/\alpha$, which is just the condition number of the function.  This is a very standard concept in convex optimization but is never referenced.

After Author Response and Reviewer Discussion: Thanks for the clarifications!  I do believe that this paper could be important in driving further research and have updated my score accordingly.

I am a bit confused about a few points in the author response though. In terms of fraction of corruptions, the paper assumes that the learner is given a starting point within some radius $R_0$ of the optimum $x^*$ (which is of course a fine and standard assumption).  Thus, the learner is optimizing over a bounded set.  My point was that almost all of the points in this set are actually uncorrupted and even within any ball of radius $2K$, almost all of the points are uncorrupted.

Also in terms of the comment about $ \log (\beta/\alpha) \leq d$, I'm not sure I understand the comparison to gradient descent needing $\beta/\alpha \leq poly(d)$. The algorithm presented in this paper still needs $d poly( \beta/\alpha)$ runtime so it needs the stronger condition as well to run in polynomial time in high dimensions.  Moreover, gradient descent works in any dimension, even low dimensions such as $d = 1$, whereas the algorithm presented here actually needs the dimension to be high.

**Time Spent Reviewing:**

2

---

> ### Author Response · Authors · 2021-08-10
> **Response to Reviewer E9DP**
>
> We thank you for your suggestions on our write-up. By “a ball centered at point $x$ of radius $R$ traps point $y$” we mean that the euclidean distance between $x$ and $y$ is at most $R$. We will use the second (more standard) terminology in our algorithm in the final version. Likewise, we will also mention that the quantity $\beta/\alpha$ is what is referred to in the literature as the condition number of the convex function.
>
> We also want to highlight one of our motivations for focusing on the high-dimensional case. Specifically, our condition that $d\geq\Omega(\log(\beta/\alpha))$ is equivalent to that the function’s condition number, $\beta/\alpha$, is at most $2^{O(d)}$. In other words, the function’s condition number is bounded by some quantity exponential in the dimension. This is the case for a wide range of convex optimization problems that we encounter in practice. In fact, in many applications, one makes a much stronger assumption: namely, that the condition number is bounded by a polynomial in $d$. As an example, first order methods such as gradient descent terminate in polynomial time if and only if the condition number $\beta/\alpha$ is at most $\mathrm{poly}(d)$.  Also, we note that the bound on the condition number is only required for the correctness of the second phase of our algorithm; the first phase does not require that.
>
> We would also like to address your concern about the fraction of corruption. First, we do not think it is meaningful to measure the “fraction” of corruption compared to the domain in our setting. In particular, since we are focused entirely on unconstrained convex optimization, an algorithm has access to all points in the infinitely large domain $\mathbb{R}^{d}$, and any of these points may potentially be helpful in locating the minimizer. We also note that a higher dimension does not make the power of corruption weaker. In fact, our lower bound shows that especially in the high dimensional regime, the adversary has the power to prevent any algorithm from approximating the minimizer to better than within distance $K\sqrt{\beta/\alpha}$, by only using an outlier noise of radius $K$. Our algorithm then shows that we are actually able to achieve this best possible approximation.

---

### Official Review · Reviewer_CLVB · 2021-07-14

**Rating:** 6
**Confidence:** 4

**Summary:**

The authors consider a convex minimization problem with a certain noisy oracle. Specifically, the function to be minimized is assumed to be alpha-strongly convex, and beta-Lipschitz (referred to as alpha-beta nice). It is also assumed that we have access to the function values everywhere except for an ellipsoid with bounded volume. Within the mentioned ellipsoid, the oracle may select the values "adversarially" so as to confuse the minimization algorithm. In this setting, the authors first show the existence of two alpha-beta nice functions that differ on a bounded ball and such that their minimizers are a certain distance apart. This provides a bound on what one could possibly achieve. Next, they provide two algorithms that guarantee to produce approximate minimizers that are sufficiently close to the minimizer (within the bounds permitted by the first negative result), along with an upper bound on the number of queries that the algorithms make.

**Main Review:**

The paper is a purely theoretical paper that delivers what it promises. It is clearly written, and well-presented, with detailed proofs in an appendix. I think some of the intuition is hidden in the proofs, but that doesn't make the paper unreadable. There are no experiments to demonstrate how the framework could be useful in practice, and I think that's a weakness of the paper. While the authors place their noise model within the context of others' works, I think a stronger motivation is lacking -- especially since the whole framework (e.g., the selection of the family of functions to work with) relies on it. Overall, I think this paper would be of high interest to a few readers -- and I'd welcome authors' comments that may serve to broaden the readership.

I have only a few minor comments :
- In presenting the model (starting line 54), the description reads as if there's no restriction on C, other than its volume. It becomes clear later that we also need C to be contained in an ellipsoid -- in fact, otherwise the oracle could have simply chosen to corrupt each query, adding up to a total of zero volume, since there are only a finite queries. Please clarify this part.

- About Thm 3.1 : While I don't have any objection to this statement, I found it a bit pessimistic. How about a statement in the following form : "Given an arbitrary (alpha-beta) nice f, there exists (alpha-beta) g such that f and g differ only on a set contained in a ball of radius R.". Does such a statement follow from your result? If not, is it possible that some functions are "harder" for the oracle to scramble?

- In the appendix, Remark D.1 mentions that, for simplicity, the gradients will be assumed to be exact, with a certain probability. Does this really not affect any of the coming proofs? When you work with a bounded error in the gradient (again with the same small probability), if the proofs extend easily, why do you make this assumption? The appendix is already detailed, so you might as well drop this assumption. If there's a compelling reason to keep it, I'd suggest to make it part of the statements of the results.

-----------------------------------------
Update after author responses :

I read the other reviews and the authors' responses.
Generally, I agree that the model considered is a bit unusual, but I think the manuscript is potentially interesting for a small community, and might spark a more useful discussion. I'm still somewhat positive, and will keep my score as is.

**Time Spent Reviewing:**

8

---

> ### Author Response · Authors · 2021-08-10
> **Response to Reviewer CLVB**
>
> We thank you for your valuable comments. First we would like to briefly comment on the likely audience for our paper. Aside from the noisy optimization community mentioned in our introduction section, we believe our paper will also be of interest to the rich community of robust learning, in which the outlier noise (or equivalently, $\ell_0$ bounded noise) is one of the challenging models for data corruption. Some relevant works from this community are mentioned in Appendix A in our paper.
>
> We address below your other comments in some detail:
>
> - There is indeed no restriction on the corrupted set $C$ other than its volume. In particular, $C$ does not have to be contained in an ellipsoid. On the other hand, note that in our lower bound, the set $C$ is indeed an ellipsoid. The high-level message is that the best strategy for the adversary to fool the algorithm is to cluster all the corruption together. However, the algorithm does not put any restriction on the shape of $C$.
> While we allow the shape of $C$ to be arbitrary, we note that (as in previous works on noisy optimization) in our model, the corruption set $C$ is chosen beforehand and is oblivious to the algorithm. As the query points are chosen randomly, it is unlikely that all of them lie in $C$, thus thwarting the kind of attack you mentioned.
>
> - We do not immediately see how our Thm 3.1 implies a similar statement for any $(\alpha,\beta)$-nice function, as the convex interpolation conditions are somewhat subtle. We think it is certainly possible that some functions are harder for the adversary to scramble. What we show is that the outlier noise model is so powerful that even for a simple quadratic function - namely  the squared $\ell_2$ norm - it is impossible to find the minimizer better than what is guaranteed by our upper bounds.
>
> - The reason we assume the gradients are exact is to make the proof cleaner and easier to verify by the reader, as otherwise the proof will involve a lot of applications of triangle inequality and the error analysis will become messier. However, we do agree that adding a proof with approximate gradients will make our paper more thorough. We will do this in the final version of our paper.

---

### Official Review · Reviewer_jGRm · 2021-07-27

**Rating:** 6
**Confidence:** 3

**Summary:**

The paper studies the question of optimizing a strongly convex function f given a zeroth' order oracle for the function. This is one of the most studied problems in optimization and there is a wealth of literature on the problem.

The twist considered here is that the zeroth-order oracle has outlier noise: on some points in the domain, the oracle can output an arbitrary value (completely corrupted). The fraction of the domain that is corrupted is assumed to have a volume equal to that of a ball of radius K.

Clearly, since we are allowed to completely corrupt a ball of radius K, one cannot expect to get more than K close to the true minimizer even with an unbounded number of queries. Interestingly, the authors also show that if the function f is alpha-strongly convex and beta-smooth, one, in fact, cannot even get better than $K \sqrt{\beta/\alpha}$ close with an unbounded number of queries. The paper also shows that one can get relatively close: If the function f is alpha-strongly convex and beta-smooth, then one can get to a point that is $O(K \sqrt{\beta/\alpha})$ close and this with $\approx O(d (\beta/\alpha)^2)$ queries.

The model and results are nice. One drawback though is that the results only apply to high dimensional scenarios.

**Limitations And Societal Impact:**

Yes

**Main Review:**

Lower bound: The lower bound that one cannot get better than $O(K \sqrt{\beta/\alpha})$ is obtained by explicitly constructing two functions whose minima are $K\sqrt{\beta/\alpha}$ apart but differ from each other only on a ball of radius $K$. This critically relies on the space being high-dimensional.

Upper bound: The authors first use plain gradient descent to argue that one can indeed get $O(K \beta/\alpha)$ close relatively easily. The authors then bootstrap by a clever argument of averaging the gradients in a ball of radius $2K$. Once again, as we are in high dimensions, the fractional volume covered by a ball of radius $K$ in a ball of radius $2K$ is exponentially small and as such most points will give the correct answer.

**Time Spent Reviewing:**

3

---

> ### Author Response · Authors · 2021-08-10
> **Response to Reviewer jGRm**
>
> We thank you for your positive feedback. We only want to briefly remark that our condition on the dimension is actually very mild. Specifically, our condition that $d\geq\Omega(\log(\beta/\alpha))$ is equivalent to that the function’s condition number, $\beta/\alpha$, is at most $2^{O(d)}$. In other words, the function’s condition number is bounded by some quantity exponential in the dimension. This is the case for a wide range of convex optimization problems that we encounter in practice. In fact, in many applications, one makes a much stronger assumption: namely, that the condition number is bounded by a polynomial in $d$. As an example, first order methods such as gradient descent terminate in polynomial time if and only if the condition number $\beta/\alpha$ is at most $\mathrm{poly}(d)$.  Also, we note that the bound on the condition number is only required for the correctness of the second phase of our algorithm; the first phase does not require that.

---

### Decision · Program_Chairs · 2021-09-27

**Decision:**

Accept (Poster)

**Comment:**

This paper studies the problem of minimizing a convex function given an evaluation oracle to it -- with the twist that the oracle has outlier-noise: for some parameter K, a volume equal to that of a ball of radius K can be corrupted adversarially (thus, when evaluated at any point inside this ball, the oracle can return an arbitrary, potentially adversarial noise). This immediately puts a lower bound of K on the distance to the optimum. The paper proves a non-trivial lower bound of K\sqrt{\beta/\alpha} on the best distance guarantees to the optimum for any \alpha-convex \beta-smooth function and shows that one can achieve this bound within a constant factor with poly(\beta/\alpha) queries.

The idea of the algorithm is simple and natural (get somewhat close to the optimum by gradient descent and then do a clever bootstrapping by averaging gradients in a ball). But the paper studies and proves a clean result about a basic problem of interest to optimization and machine learning community broadly construed. We recommend acceptance.